# Electrical switching of Ising-superconducting nonreciprocity for quantum neuronal transistor

Junlin Xiong [1,5], Jiao Xie[1,5], Bin Cheng [2] ✉, Yudi Dai[1], Xinyu Cui[1], Lizheng Wang [1], Zenglin Liu[1], Ji Zhou[1], Naizhou Wang [3], Xianghan Xu [4], Xianhui Chen [3], Sang-Wook Cheong [4], Shi-Jun Liang [1] ✉ & Feng Miao [1] ✉

Nonreciprocal quantum transport effect is mainly governed by the symmetry breaking of the material systems and is gaining extensive attention in condensed matter physics. Realizing electrical switching of the polarity of the nonreciprocal transport without external magnetic field is essential to the development of nonreciprocal quantum devices. However, electrical switching of superconducting nonreciprocity remains yet to be achieved. Here, we report the observation of field-free electrical switching of nonreciprocal Ising superconductivity in $Fe_3GeTe_2/NbSe_2$ van der Waals (vdW) heterostructure. By taking advantage of this electrically switchable superconducting nonreciprocity, we demonstrate a proof-of-concept nonreciprocal quantum neuronal transistor, which allows for implementing the XOR logic gate and faithfully emulating biological functionality of a cortical neuron in the brain. Our work provides a promising pathway to realize field-free and electrically switchable nonreciprocity of quantum transport and demonstrate its potential in exploring neuromorphic quantum devices with both functionality and performance beyond the traditional devices.

Nonreciprocity is an inherent property of materials describing the inequality of the electric or optical signals traveling in opposite directions. This nonreciprocity is usually strongly correlated to the intricate interplay among various types of symmetry breaking, such as inversion symmetry, time-reversal symmetry and mirror symmetry[1–7]. Manipulation of the nonreciprocity by switching the polarized states resulting from the spontaneous symmetry breaking could give unique opportunities for developing future technologies. Particular interest in this field lies in the nonreciprocal superconducting transport[8–18], which could be exploited to develop next-generation electronic devices that have ultralow dissipation and new functionalities, such as electronic

synapses and neurons[19–25]. A key step for that purpose is achieving electrical switching of the nonreciprocal superconductivity without the assistance of the external magnetic field, but the electrically switchable nonreciprocal superconductivity is still absent. It's worth noting that the polarity of such a superconducting diode effect is usually strongly entwined with the breaking mechanisms of time-reversal symmetry and crystalline symmetry[26]. To that end, searching unconventional superconductors with magnetization- or ferroelectricity- determined nonreciprocity would give a unique opportunity for electrically switching the superconducting diode behavior by reversing those polarized symmetry-broken states[13,27].

[1]Institute of Brain-Inspired Intelligence, National Laboratory of Solid State Microstructures, School of Physics, Collaborative Innovation Center of Advanced Microstructures, Nanjing University, 210093 Nanjing, China. [2]Institute of Interdisciplinary Physical Sciences, School of Science, Nanjing University of Science and Technology, 210094 Nanjing, China. [3]Hefei National Laboratory for Physical Science at Microscale and Department of Physics and Key Laboratory of Strongly Coupled Quantum Matter Physics, University of Science and Technology of China, 230026 Hefei, Anhui, China. [4]Center for Quantum Materials Synthesis and Department of Physics and Astronomy, Rutgers, The State University of New Jersey, Piscataway, NJ 08854, USA. [5]These authors contributed equally: Junlin Xiong, Jiao Xie. ✉e-mail: bincheng@njust.edu.cn; sjliang@nju.edu.cn; miao@nju.edu.cn

In this work, we demonstrate the electrically switchable non-reciprocal Ising superconductivity at zero magnetic field in a perpendicular-anisotropy Ising-superconducting (PAIS) quantum material. The PAIS material is synthesized by stacking a layered Ising superconductor, which has Ising-type spin-orbit coupling (SOC)[28], onto a vdW magnet of perpendicular anisotropy[29]. With the high-quality vdW interface, strong magnetic proximity is induced in the Ising superconductor and breaks time-reversal symmetry, leading to the emergence of field-free nonreciprocal Ising-superconductivity, i.e., superconducting diode effect. Moreover, the polarity of nonreciprocal superconductivity in the PAIS device can be switched by electrically reversing the magnetization at zero magnetic field through current-induced out-of-plane spin accumulation. Based on the electrical manipulation of this magnetization-determined superconducting diode effect, the proposed nonreciprocal neuronal transistor can implement an XOR logic gate and faithfully emulate the biological functionality of a cortical neuron in the brain. Our work demonstrates that the electrical switching of nonreciprocal quantum transport in condensed matter systems shows great potential in neuromorphic computing.

## Results

### Field-free and magnetization-determined superconducting diode

The PAIS device was fabricated by stacking vdW magnet $Fe_3GeTe_2$ (FGT) and Ising superconductor 2H-NbSe$_2$ (Fig. 1a, b). We carried out measurements of the longitudinal resistance under different temperatures. A typical PAIS device with five-layer NbSe$_2$ (see the atomic force microscope images and corresponding height profiles in Supplementary Fig. 1) exhibits a superconducting behavior with a transition temperature of $T_c \approx 2.95$ K (Fig. 1c). To characterize the magnetization of the PAIS material, we measured the Hall resistance under various magnetic field (Supplementary Fig. 2). The results show hysteresis loops of Hall resistance which shrink as temperature increases, indicating that the PAIS device possesses switchable perpendicular magnetization states, denoted as magnetization "UP" and "DOWN" states.

We then investigated the superconducting behaviors under different magnetization states. We first set the magnetization state of the device as "UP" and swept the d.c. current under different perpendicular magnetic fields $B_z$, and monitored the longitudinal resistance at the temperature of 1.6 K. As shown in Fig. 1d, sharp jumps of resistance

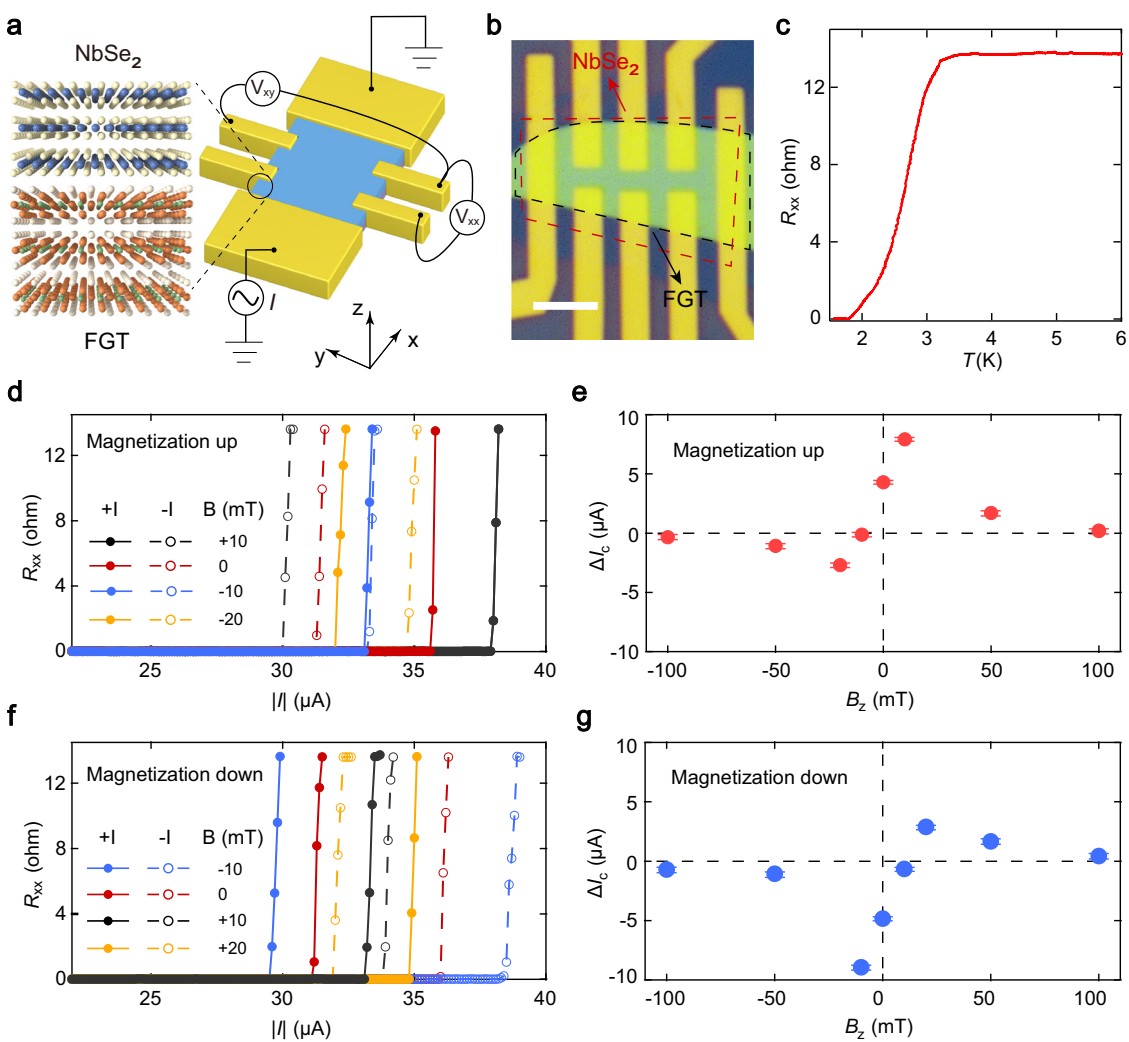

**Fig. 1 | Measurement configuration and magnetization-determined non-reciprocal Ising superconductivity. a** Schematic of the PAIS device consisting of NbSe$_2$ and FGT flakes for electrical transport measurements. **b** Optical image of a typical device. The scale bar is 3 μm. **c** Temperature dependence of the device resistance with an applied electrical current of 0.5 μA. **d** Current dependences of the resistance under various magnetic fields for both positive and negative currents at 1.6 K when the magnetization is set as "UP" state. **e** The nonreciprocal component of the critical current $\Delta I_c$ as a function of the magnetic field for the magnetization "UP" state. **f** Current dependences of the resistance under various magnetic fields for both positive and negative currents at 1.6 K when the magnetization is set as "DOWN" state. **g** The nonreciprocal component of the critical current $\Delta I_c$ as a function of the magnetic field for the magnetization "DOWN" state.

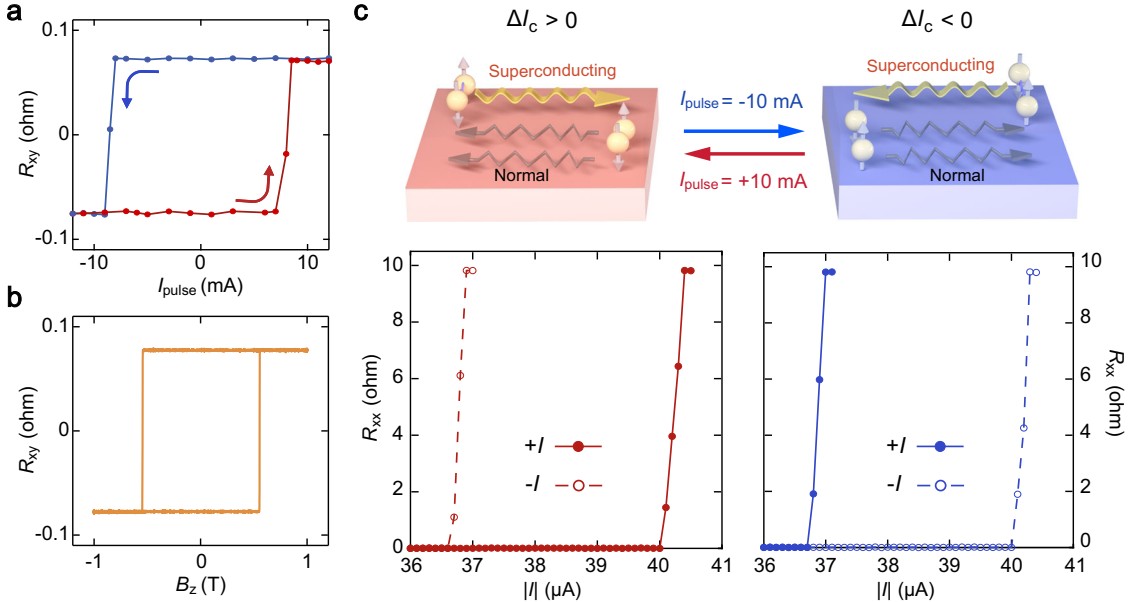

**Fig. 2 | Electrically switchable nonreciprocal Ising superconductivity. a** Current-induced magnetization switching at zero field at 1.6 K. The sign of Hall resistance is reversed by sweeping the pulsed current. **b** Hall resistance as a function of the perpendicular magnetic field at 1.6 K. The Hall resistance was measured using an a.c. excitation current of 500 μA. **c** Schematics (upper panels) and experimental data (lower panels) of electrically switchable nonreciprocal superconducting transport. After applying a positive current pulse ($I_{pulse}$ = +10mA), the polarity of nonreciprocal superconductivity is switched to $\Delta I_c$>0, i.e., nonreciprocal supercurrent flowing rightward (left panels). While the negative current pulse ($I_{pulse}$ = − 10mA) favors the reversal to the polarity "-" state, i.e., $\Delta I_c$<0 (right panels).

occur as the current reaches a critical value ($|I_c|$). When $B_z$ is applied in the positive direction of the z-axis (e.g., $B_z$ = +10mT), $|I_c|$ strongly depends on the direction of the current flowing (positive +$I$ or negative −$I$), indicating the nonreciprocity of the supercurrent, i.e., superconducting diode effect. Such nonreciprocal supercurrent, characterized by $\Delta I_c = |I_c^+| − |I_c^-|$, is dependent on the applied magnetic field, as shown in Fig. 1e. Here, $\Delta I_c$>0 and $\Delta I_c$<0 corresponds to the "+" and "−" polarity, respectively. Notably, as $B_z$ is set to zero, a significant nonreciprocal supercurrent with "+" polarity still exists, suggesting the emergence of a superconducting diode effect does not require external field. This nonreciprocity with "+" polarity is retained until $B_z$ = − 10mT and eventually reversed to that with "−" polarity (i.e., $\Delta I_c$<0). When the magnetic field further increases, the nonreciprocal supercurrent $\Delta I_c$ is then suppressed beyond a certain magnetic field due to field-induced breakdown of superconductivity (Supplementary Fig. 5). In contrast, we observed the nonreciprocal supercurrent with the same magnitude but opposite polarity, when the magnetization is switched to the "DOWN" state (Fig. 1f, g). Such nonreciprocity of superconducting transport can persist to zero external magnetic field in our PAIS device. This is in stark contrast to that reported in nonmagnetic superconducting device[8], in which the nonreciprocal superconducting transport only emerges at non-zero external magnetic field. Such zero-field superconducting diode effect can produce a large nonreciprocal efficiency, defined by $\eta = \frac{2(|I_c^+|-|I_c^-|)}{|I_c^+|+|I_c^-|}$, up to 13% at 1.6 K, which is much larger than the value (6% at 0.9 K) previously reported in NbSe$_2$-based heterostructure[16]. Since the direction of electrical transport is perpendicular to the magnetization, the observed nonreciprocal superconductivity is similar to magneto-toroidal nonreciprocal directional dichroism (NDD) effect[5]. Note that the nonreciprocal superconducting behavior is determined by the magnetization state in our PAIS device. Electrical switching of the magnetization is thus critical for developing ultralow-power electronic devices based on the superconducting diode effect.

## Field-free electrical switching of superconducting diode
We demonstrated that such electrical switching of the polarity of nonreciprocal superconductivity in the PAIS device can be realized through d.c. current pulse. We first swept the pulsed d.c. current $I_{pulse}$ applied to the PAIS device at 1.6 K and monitored the change in the Hall resistance, with the corresponding results presented in Fig. 2a. At zero magnetic field, sweeping $I_{pulse}$ upwards or downwards to a critical value can change the sign of the Hall resistance. This phenomenon is consistent with the anomalous Hall effect observed in the same device (Fig. 2b), confirming that the d.c. current pulse can switch the magnetization state in the PAIS device. To be specific, a positive current pulse favors the reversal of the magnetization to "UP" state, while a negative current pulse switches the magnetization to "DOWN" state. Such field-free electrical switching of magnetization could enable the electrical manipulation of nonreciprocal superconducting transport due to the intrinsic intertwinement of magnetization and nonreciprocal superconductivity. As shown in Fig. 2c, after applying a positive current pulse ($I_{pulse}$ =+10 mA), we observed the field-free nonreciprocal superconductivity with "+" polarity, consistent with the behavior of magnetization "UP" state demonstrated in Fig. 1d. By contrast, a negative current pulse ($I_{pulse}$ =−10 mA) can switch the nonreciprocal superconductivity with "+" polarity to its opposite, which is consistent with the behavior of magnetization "DOWN" state shown in Fig. 1f. Such electrical switching of the nonreciprocal Ising superconductivity could provide a promising pathway for the development of novel nonreciprocal electronic devices[17].

## Origin of field-free electrically switchable nonreciprocal superconductivity
We infer that the nonreciprocal Ising superconductivity could result from the intricate interplay among the valley-contrasting Ising spin-orbit coupling (SOC) effect, the time-reversal symmetry breaking induced by magnetic proximity effect and the lowered rotation symmetry in the PAIS quantum materials. On the one hand, the ubiquitous

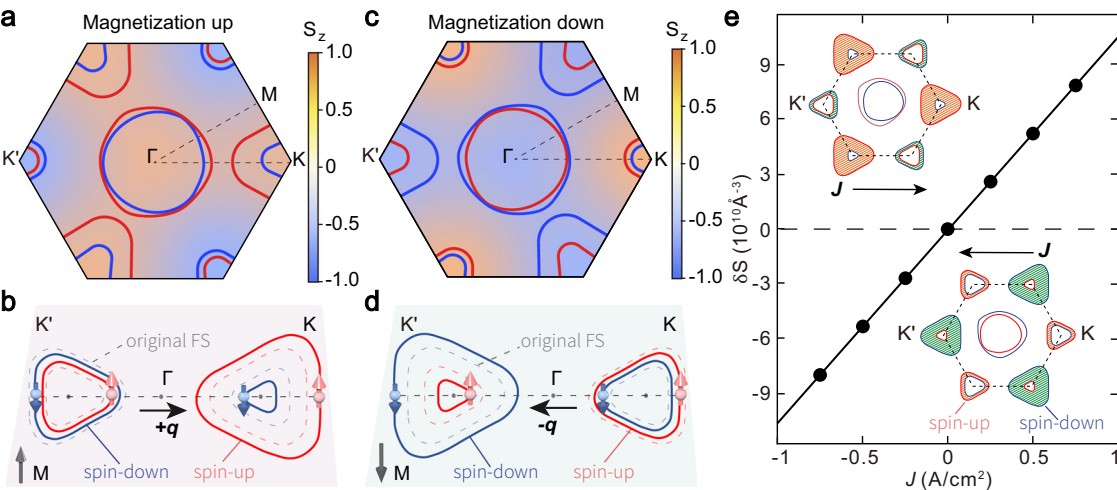

**Fig. 3 | The mechanism for electrically switchable superconducting diode. a** Fermi surface and spin texture of NbSe$_2$ based on the tight-binding Hamiltonian in the first Brillouin zone for magnetization "UP" state. The color scale indicates the out-of-plane spin component. The red and blue lines represent the energy bands with the spin "UP" and "DOWN" states, respectively. **b** Schematic of magnetic proximity induced finite momentum of Cooper pairs for magnetization "UP" state. The dotted and solid lines represent the energy bands without and with the magnetic proximity effect, respectively. **c** Fermi surface and spin texture of NbSe$_2$ for magnetization "DOWN" state. **d** Schematic of magnetic proximity induced a finite momentum of Cooper pairs for magnetization "DOWN" state. **e** Calculation of current-induced spin density. Insets are the schematic illustrations of spin distributions changed by the current-induced electric fields. Red and blue lines represent contours of spin-up and spin-down branches, respectively. Orange and green shaded regions represent spin-up and spin-down overpopulation for K and K' valleys, respectively.

strain and lattice mismatch at the vdW interface can break the $\mathcal{M}_y$ mirror symmetry of the NbSe$_2$, and lead to in-plane electrical polarization (**P**). This in-plane **P** and perpendicular magnetization **M** of PAIS device can form a magneto-toroidal moment[30], leading to magneto-toroidal NDD with nonreciprocal charge transport. When charge current is applied, such lowered lattice symmetry at the interface can give rise to valley-asymmetric Berry curvature distribution, and thus lead to valley magnetization with perpendicular anisotropy, which can generate a spin torque and facilitate the switching of magnetization in PAIS device[31]. In this case, the magneto-toroidal NDD can be flipped by electric pulse through the flipping of **M** with **P** fixed.

On the other hand, with the current along the zigzag direction, the $\mathcal{M}_y$ mirror symmetry of this system can also be broken due to the intricate interplay between valley-contrasting triagonal warping and the magnetic proximity effect (see more detailed discussion in Supplementary Materials). As shown in Fig. 3a, b, an upward magnetization enhances the spin polarization in K valley while decreases the spin polarization in K' valley. In conjunction with the asymmetric trigonal warping effects present in the two distinct valleys, a finite momentum of Cooper pairs could emerge in NbSe$_2$. Conversely, a downward magnetization gives rise to the non-zero Cooper pairs momentum in an opposite manner (Fig. 3c, d). Notably, such effect is a quadratic effect compared to the fermi energy which dominates the charge transport properties. However, the superconducting gap is several orders smaller than the fermi energy, leading to prominent non-reciprocal quantum transport behavior near the superconducting region where superconducting gap is the dominant energy scale. Such understanding can be further verified by our successful observation of prominent second harmonic behavior near the superconducting transition region (see Supplementary Fig. 6 and Methods for details), which decays fast when the superconductivity is fully suppressed by external magnetic field and the fermi energy becomes the dominate energy scale. To further clarify this mechanism, we employ the generalized Ginzburg-Landau (GL) theory[3,32] in the proximity to the transition temperature $T_c$ (see methods for details). When the magnetic proximity and trigonal warping effect are considered in the Ising superconductor NbSe$_2$, we demonstrate that the nonreciprocity of superconducting transport is determined by both the external

magnetic field $B_z$ and the proximity-induced magnetization $M_z$, which is consistent with our experimental results shown in Fig. 2. In addition, the nonreciprocal efficiency $\eta$ is calculated to be proportional to $(T_c - T)^{1/2}$, which coincides with our experimental result shown in Supplementary Fig. 7 (see details in Methods). Finally, the calculation shows that the valley-contrasting Ising SOC with triagonal warping effect could also facilitate the generation of current-induced perpendicular spin polarization (see Methods for details), as shown in the inset of Fig. 3e. This current-induced z-spin could also produce a spin torque at the vdW interface and thus contribute to the switching of the magnetization in the PAIS material (Supplementary Fig. 8). It is noted that this electrically switchable nonreciprocal superconductivity is reproducible and observed in the both PAIS devices with odd-layer and even-layer NbSe$_2$ (see Section IX of the Supplementary Materials). Fully understanding of such electrical switching of superconducting nonreciprocity requires microscopic models considering the details of the magnetic proximity effect and the symmetry breaking at the vdW interface[27,33–35].

## Quantum neuronal transistor based on electrically switchable superconducting diode effect

Taking advantage of the electrically switchable nonreciprocal Ising superconductivity, we propose and demonstrate a proof-of-concept quantum neuronal transistor, which can faithfully emulate the biological functionality of a cortical neuron[36] in the brain, i.e., performing nonlinear computing operations (Fig. 4a). The neural transistor, a tetrode device shown in Fig. 4b, is operated by feeding input and control signals (represented by X and Y) into the NbSe$_2$ film through the electrodes. The control current pulse signal Y is used for deterministic reversal of the perpendicular magnetization and thus determines the polarity state ("$\Delta I_c > 0$" and "$\Delta I_c < 0$" corresponds to state "1" and "0", respectively) of nonreciprocal superconducting transport. The resulting state would generate a distinct electrical output corresponding to the input current signal X. The output state ("1" or "0") is represented by the resistance (high or low). To demonstrate the function of the proposed transistor, we switched the polarity state of nonreciprocal superconductivity by applying a train of positive and negative current pulses (denoted as $I_{switch}$) onto this device (Fig. 4c)

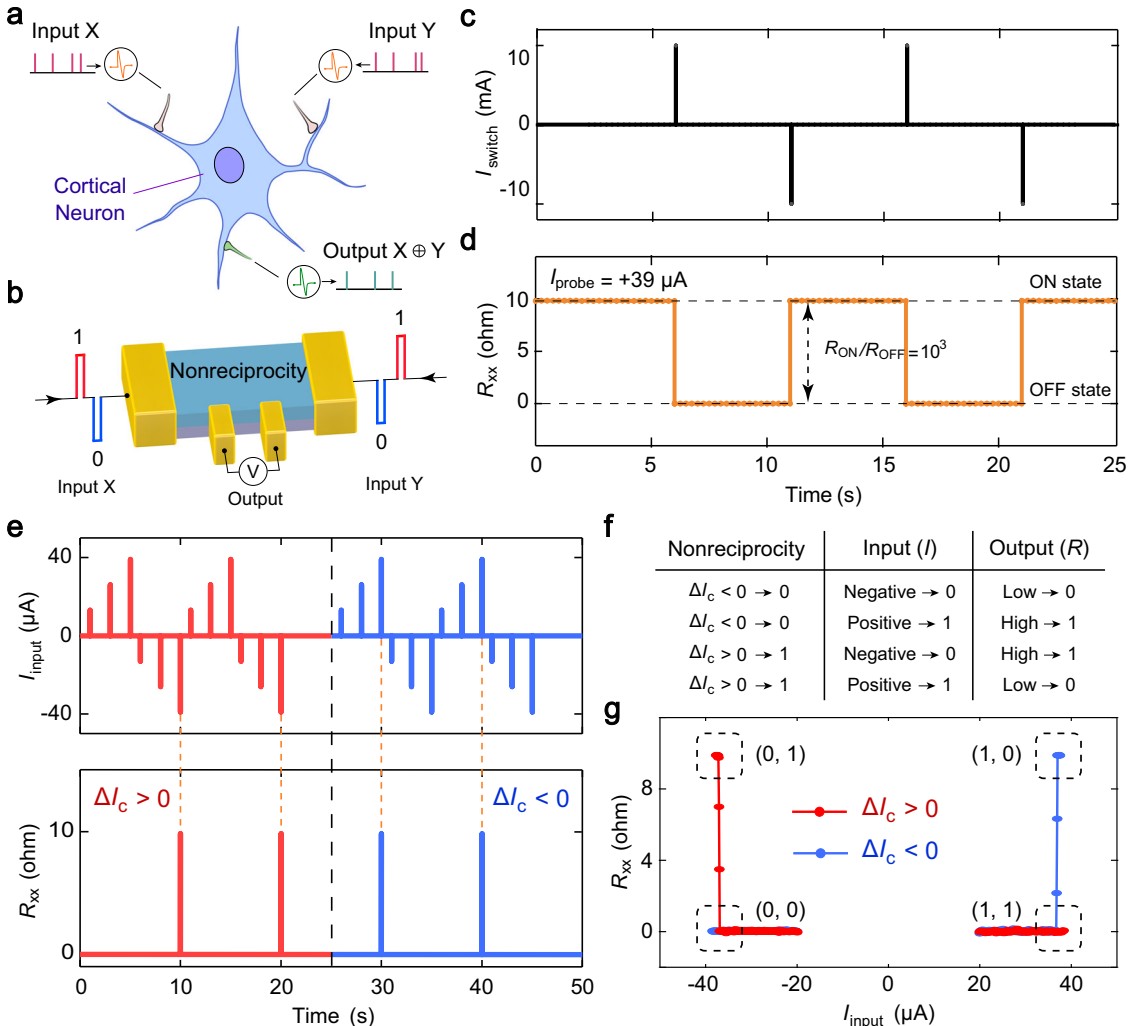

**Fig. 4 | Nonreciprocal neural transistor. a** Schematic structure of a biological cortical neuron. The neuron can classify linearly nonseparable inputs through the nonlinear XOR function. **b** Schematic of the neural transistor. Input signals (represented by X and Y) are fed into the NbSe$_2$ film through the electrodes. **c** Deterministic switching by a series of current pulses applied in the device. The width and magnitude of the current pulses are 200 μs and 10 mA, respectively.

**d** The resistance is measured by using a small d.c. excitation current of +39 μA. **e** The responses of spike to the input current pulses for the polarity "+" and "−" states. **f** Logic function of the neural transistor, in which nonlinear input-output responses depend on the polarity state. **g** The XOR function in the nonreciprocal neural transistor. The dashed boxes represent the logic state values for input and polarity combinations (0,1), (1,1), (0,0), and (1,0), respectively.

and measured the resulting output (Fig. 4d). For the probe current $I_{probe}$ of +39 μA, the ON-state resistance ($R_{ON}$) corresponding to the polarity "+" state is smaller than 0.004 Ω, whereas the OFF-state resistance ($R_{OFF}$) corresponding to the polarity "-" state is about 9.8 Ω, giving rise to an on/off ratio of $10^3$. This value is two orders of magnitude larger than that of MgO-based conventional MTJs[37–39] and one order of magnitude larger than the state-of-the-art value (19,000%) under similar experimental conditions[40]. Notably, this ratio depends on the lowest resolution of the measurement system and can be further improved by improving the detection precision. We also note that a superconducting transistor based on the distinct physical mechanism has been theoretically proposed in a Josephson junction with chiral magnet[41]. Unlike the Josephson transistor based on the junction structure in that work, our proposed superconducting neuronal transistor is realizable in all-metallic junction-free superconductors.

The quantum neuronal transistor can emulate the biological function of a cortical neuron[36] in the brain, which can classify linearly nonseparable inputs. As the polarity of nonreciprocity is in "+" state, the transistor exhibits a threshold response behavior due to current induced transition between the superconducting and normal states, and only spikes (i.e. output jumps from state "0" to state "1" and back to state "0")

when receiving negative current pulses of large magnitude (Fig. 4e). In contrast, the transistor with polarity "-" state only spikes when receiving positive current pulses of large magnitude. These threshold response behaviors resemble those features in the cortical neurons capable of executing XOR nonlinear computational functions. Moreover, we show that the neuronal transistor can also realize the function of XOR gate (Fig. 4f). To demonstrate the XOR function, we set the positive current pulse (+39 μA) as input logic state "1" and the negative current pulse (−39 μA) as input logic state "0" in the experiment. As shown in Fig. 4g, when the input state and polarity state are set to (0,1) and (1,0), logic state "1" corresponding to a high resistance can be generated. By contrast, logic state "0" corresponding to a low resistance can be output when the input state and polarity state are set to (0,0) and (1,1). These results indicate that an XOR gate can be realized in a single quantum transistor. Note that this nonlinear Boole logic function cannot be implemented with a single traditional device and is conventionally thought to require multilayered networks[42–44]. In addition, the PAIS device allows the simultaneous achievement of the giant on/off ratio (>200,000%) and ultralow resistance area product (≈0.1 Ω μm$^2$), which cannot be realized in conventional MTJs (see detailed comparison in Supplementary Fig. 12) but urgently required for ultrahigh-density electronic applications[45,46].

## Discussion

In summary, we demonstrate field-free electrical switching of Ising superconducting diode effect in the Fe₃GeTe₂/NbSe₂ van der Waals (vdW) heterostructure. By taking advantage of this electrically switchable superconducting diode effect, we propose and demonstrate a nonreciprocal quantum neuronal transistor able to perform the XOR function, which is inaccessible with previously reported technology. Our work opens up a promising avenue for neuromorphic computing based on nonreciprocal quantum transport.

## Methods

### Device fabrication and fundamental characterization

We mechanically exfoliated the $NbSe_2$ and FGT flakes onto a highly doped Si wafer covered by a 300-nm-thick $SiO_2$ layer. Before the device fabrication, the crystallographic orientation of the $NbSe_2$ flake was characterized by measuring the co-polarized SHG intensity as a function of the relative angle between laser polarization and crystal orientation. In the following fabrication process, we designed the device geometry to set the direction of current along the zigzag direction of the $NbSe_2$ sample in the experiments. As shown in Supplementary Fig. 14, the maximum (minimum) intensity corresponds to the armchair (zigzag) direction of the crystal, confirming the current flowing along the zigzag direction. The thickness of these flakes was identified with optical contrast and a Bruker MultiMode 8 atomic force microscope. The $NbSe_2$ and FGT flakes of a typical PAIS device are approximately 3.1 nm (5 layers) and 28.9 nm in thickness, respectively (see the atomic force microscope images and corresponding height profiles in Supplementary Fig. 1). The bottom electrodes (2 nm Ti/ 30 nm Au) were patterned using the standard electron beam lithography method and deposited by standard electron beam evaporation. Poly(propylene) carbonate (PPC) coated on polydimethylsiloxane (PDMS) was used to pick up the $NbSe_2$ and FGT flakes and fabricate the heterostructure devices in a glovebox filled with an inert atmosphere to avoid degradation.

### Electrical measurements

All the electrical measurements were performed in the Oxford cryostat with magnetic fields of up to 14 T and temperatures between 1.5 and 300 K. To characterize the electric transport state, a Keithley 2636B dual-channel digital source meter or lock-in amplifier (Stanford SR830) was used to inject the d.c. or a.c. current and measure the 4-probe resistance. The a.c. measurements were performed by injecting a.c. current with a frequency of $\omega$ (17.777 Hz) using lock-in amplifiers (Stanford SR830). In the measurement of electrical switching of magnetization, large current pulses (write current, 200 μs) were first applied by using Keithley 2636B dual-channel digital source meter. After an interval of 5 s, the Hall resistance was measured using an alternating current excitation current of 500 μA.

### Effective tight-binding Hamiltonian

For a $NbSe_2$/FGT heterostructure, the tight-binding Hamiltonian[47–50] can be given by

$$H(\boldsymbol{k}) = \varepsilon(\boldsymbol{k}) + h_{Ising}(\boldsymbol{k}) + h_{proximity} \tag{1}$$

where $\varepsilon(\boldsymbol{k}) = -\sum_{j=1}^{3}\{2t_1\cos k_j + 2t_2\cos(k_j - k_{j+1})\} - \mu$ is the kinetic energy with $k_j = \boldsymbol{k}\cdot\boldsymbol{R}_j$, and the unit lattice vectors $\boldsymbol{R}_1 = \hat{\boldsymbol{y}}$, $\boldsymbol{R}_2 = -\frac{\hat{\boldsymbol{y}}}{2} - \frac{\sqrt{3}}{2}\hat{\boldsymbol{x}}$, $\boldsymbol{R}_3 = -\frac{\hat{\boldsymbol{y}}}{2} + \frac{\sqrt{3}}{2}\hat{\boldsymbol{x}}$, $\boldsymbol{R}_4 \equiv \boldsymbol{R}_1$. Here $\mu$ is the chemical potential, $t_1$ and $t_2$ denote the nearest-neighbor (NN) and next-nearest-neighbor (NNN) hoping integrals, respectively. $h_{Ising}(\boldsymbol{k}) = -\beta(\boldsymbol{k})\sigma_z$ is the energy caused by Ising spin-orbit coupling originated from inversion symmetry breaking of $NbSe_2$, where $\beta(\boldsymbol{k}) = \lambda_I(\sin k_1 + \sin k_2 + \sin k_3)$. $h_{proximity} = -M_z\sigma_z$ is the energy caused by the proximity effect from the perpendicular magnetization. We set $(t_1, t_2, \mu, \lambda_I) = (0.009, -0.093, 0, 0.025)$ eV to fit the electronic band structure from DFT calculations[49,51]. The band structure and spin texture of $NbSe_2$ for different magnetization states are shown in Fig. 3a, c in the main text. As shown in Fig. 3a, an upward magnetization enhances the spin polarization in K valley while decreases the spin polarization in K' valley, thereby lifting the valley degeneracy. Conversely, a downward magnetization can lift the valley degeneracy in a opposite manner (Fig. 3c).

### Nonreciprocal critical current originated from finite momentum

We employ the generalized Ginzburg-Landau (GL) theory[3,32] to elucidate the magnetization-determined nonreciprocal superconductivity in our PAIS device. The effective Hamiltonian of PAIS material is given by,

$$H(\boldsymbol{k}) = \varepsilon(\boldsymbol{k}) + h_{Ising}(\boldsymbol{k}) + h_{proximity} \tag{2}$$

where $\varepsilon(\boldsymbol{k}) = \frac{\hbar^2\boldsymbol{k}^2}{2m} - \mu$ is the kinetic energy with $\boldsymbol{k} = (k_x, k_y)$, $h_{Ising}(\boldsymbol{k}) = \lambda_I\sigma_3 + \gamma k_x(k_x^2 - 3k_y^2)\sigma_3$ and $h_{proximity} = H_z^{eff}\sigma_3$ are the energy of the Ising spin-orbit interaction and the magnetic proximity effect, respectively. Here, $H_z^{eff} \equiv B_z + \kappa M_z$ is defined as an effective field contributed from the external magnetic field $B_z$ and the proximity perpendicular magnetization $M_z$. Here, $\kappa$ is a constant to ensure dimensional consistency. The effective Hamiltonian $H(\boldsymbol{k})$ has the mirror symmetry $\mathcal{M}_x$ since $\mathcal{M}_x H(k)\mathcal{M}_x^{-1} = H(k)$, but breaks the mirror symmetry $\mathcal{M}_y$ since $\mathcal{M}_y H(k)\mathcal{M}_y^{-1} \neq H(k)$.

To derive the microscopic GL free energy, the mean-field Hamiltonian is given by

$$H_{MF}(\Delta, \boldsymbol{q}) = \frac{1}{2}\sum_k \Psi^\dagger(\boldsymbol{k}, \boldsymbol{q})\mathcal{H}(k, \boldsymbol{q})\Psi(\boldsymbol{k}, \boldsymbol{q}) + const, \tag{3}$$

where $\mathcal{H}(\boldsymbol{k}, \boldsymbol{q}) = \begin{pmatrix} H(\boldsymbol{k}+\boldsymbol{q}) & \Delta i\sigma_2 \\ -\Delta i\sigma_2 & -H^\tau(-\boldsymbol{k}) \end{pmatrix}$ represents the BdG Hamiltonian and $\Psi(\boldsymbol{k}, \boldsymbol{q}) = (c_{\boldsymbol{k}+\boldsymbol{q}\uparrow}, c_{\boldsymbol{k}+\boldsymbol{q}\downarrow}, c_{-\boldsymbol{k}\uparrow}^\dagger, c_{-\boldsymbol{k}\downarrow}^\dagger)^\tau$ is the Nambu spinor. Here, $\tau$ denotes transpose.

The free energy density can be expressed as a functional of the superconducting order parameter $\Delta$,

$$f_s(\Delta, \boldsymbol{q}) \equiv -T \ln \mathrm{Tr} \exp\left(-\frac{H_{MF}(\Delta, \boldsymbol{q})}{T}\right), \tag{4}$$

where $T$ is the temperature and the Boltzmann constant $k_B$ is neglected for simplicity[10]. We expanded the free energy density up to the third order of $\boldsymbol{q}$ and the first order of $\lambda_I$ and $\gamma$, that is,

$$f_s(\Delta, \boldsymbol{q}) = \left[\alpha_0 + \alpha_2 \boldsymbol{q}^2 + \alpha_3 q_x\left(q_x^2 - 3q_y^2\right)B_z^{eff}\right]|\Delta|^2 + \frac{\beta}{2}|\Delta|^4, \tag{5}$$

where $\alpha_0 = A_0(T - T_c)$, $\alpha_2 = \frac{\hbar^2}{4m}$, $\alpha_3 = A_3 \frac{\lambda_I \gamma}{T_c^2}$ and $\beta > 0$ with $A_0, A_3 > 0$ are numerical constants.

For simplicity, we express the above free energy density in a more compact form, $f_s(\Delta, \boldsymbol{q}) = \alpha(\boldsymbol{q})|\Delta|^2 + \frac{\beta}{2}|\Delta|^4$, with $\alpha(\boldsymbol{q}) = \alpha_0 + \alpha_2 \boldsymbol{q}^2 + \alpha_3 q_x(q_x^2 - 3q_y^2)B_z^{eff}$. Denoting $\theta$ as the angle between $\boldsymbol{q}$ and $q_x$ axis, i.e., $q_x = q\cos\theta$ and $q_y = q\sin\theta$, we then can obtain

$$\alpha(\boldsymbol{q}) = \alpha_0 + \alpha_2 \boldsymbol{q}^2 + \alpha_3 \boldsymbol{q}^3 \cos 3\theta B_z^{eff}. \tag{6}$$

Additionally, the order parameter can be optimized by $\frac{\partial f}{\partial|\Delta|^2} = \alpha(\boldsymbol{q}) + \beta|\Delta|^2 = 0$, i.e., $|\Delta|^2 = -\frac{\alpha(\boldsymbol{q})}{\beta}$ with $\alpha(\boldsymbol{q}) < 0$, leading to

$$f_s(\boldsymbol{q}) = -\frac{\alpha(\boldsymbol{q})^2}{2\beta}. \tag{7}$$

Notice that the momentum dependent current is given by $I(\boldsymbol{q}) = 2\frac{\partial f_s(\boldsymbol{q})}{\partial \boldsymbol{q}}$, which is equivalent to $\frac{\beta}{2}I(\boldsymbol{q}) = |\alpha(\boldsymbol{q})|\frac{\partial \alpha(\boldsymbol{q})}{\partial \boldsymbol{q}}$ according to Eq. (7). By minimizing $\alpha(\boldsymbol{q})$ over $\boldsymbol{q}$, one can find the Cooper pair momentum $\boldsymbol{q}_0$ in the equilibrium state. With current flowing along the zigzag direction, this current $I = I\hat{\boldsymbol{x}}$ would change this Cooper pair momentum to $q_x = (\boldsymbol{q} - \boldsymbol{q}_0) \cdot \hat{\boldsymbol{x}}$, then $\alpha(q_x) = \alpha_0 + \alpha_2 q_x^2 + \alpha_3 q_x^3 \hat{\boldsymbol{x}} \cdot (\boldsymbol{B}_z^{eff} \times \hat{\boldsymbol{y}})$. By minimizing $\alpha(q_x)$ over $q_x$, we find the Cooper pair momentum in the equilibrium state,

$$q_x^0 = \frac{2\alpha_0}{3\alpha_3}\hat{\boldsymbol{x}} \cdot \left(\boldsymbol{B}_z^{eff} \times \hat{\boldsymbol{y}}\right),$$

which is a direct result of the mirror symmetry $\mathcal{M}_y$ breaking. In this case, we can expand $\alpha(q_x)$ around its minimum $q_x^0$,

$$\alpha(q) = \alpha_0 + \alpha_2 \delta q^2 + \tilde{\alpha}_3 \delta q^3, \qquad (8)$$

where $\tilde{\alpha}_3 \equiv \alpha_3 B_z^{eff}$ and $\delta q$ is define as $\delta q \equiv q_x - q_x^0$. Under this condition, we find the supercurrent is

$$\frac{\beta}{2}I(q) = 2\alpha_0 \alpha_2 \delta q + 3\alpha_0 \tilde{\alpha}_3 \delta q^2 + 2\alpha_2^2 \delta q^3 + 5\alpha_1 \tilde{\alpha}_3 \delta q^4 + 3\tilde{\alpha}_3^2 \delta q^5. \qquad (9)$$

The critical momentum $\delta q_c = \sqrt{\left|\frac{\alpha_0}{3\alpha_2}\right|}$ is then given by $\partial_{\delta q}I(q)|_{\delta q = \delta q_c} = 0$. In this way, we can obtain the critical currents $I_c^{\pm}$ corresponding to $\delta q = \mp \delta q_c$ respectively,

$$\frac{\beta}{2}I_c^{\pm} = \frac{4|\alpha_0|^{\frac{3}{2}}}{9\alpha_2}\left((3\alpha_2^3)^{\frac{1}{2}} \pm \tilde{\alpha}_3|\alpha_0|^{\frac{1}{2}}\right) + o\left(|\alpha_0|^{\frac{5}{2}}\right). \qquad (10)$$

As such, the nonreciprocal efficiency is given by

$$\eta \equiv \frac{2(I_c^+ - I_c^-)}{I_c^+ + I_c^-} = \alpha_3\left(\frac{4|\alpha_0|}{3\alpha_2^3}\right)^{\frac{1}{2}}(B_z + \kappa M_z), \qquad (11)$$

showing the temperature dependence ($\eta \propto (T - T_c)^{\frac{1}{2}}$) and magnetism dependence ($\eta \propto B_z + \kappa M_z$) of nonreciprocal efficiency.

## Second harmonic signals

We adopted the well-established time-dependent GL equation, as widely used in previous reports[3,52], to calculate the second harmonic signal in our device. By introducing a uniform electric field $\boldsymbol{E}$ and setting the vector potential $\boldsymbol{A} = -\boldsymbol{E}t$, the GL free energy quadratic in the order parameter reads

$$F = \int d\boldsymbol{r}\Delta^*(\boldsymbol{r}, t)\alpha(\boldsymbol{r}, t)\Delta(\boldsymbol{r}, t) = \sum_q \alpha(\boldsymbol{q}, t)|\Delta(\boldsymbol{q}, t)|^2, \qquad (12)$$

where $\alpha(\boldsymbol{q}, t)$ is the time-dependent GL coefficient which can be expressed as

$$\begin{aligned}\alpha(\boldsymbol{q}, t) = &\, \alpha_0 + \alpha_2\left(\boldsymbol{q} - \frac{2e}{\hbar}\boldsymbol{E}t\right)^2 \\ &+ \alpha_3 H\left(q_x - \frac{2e}{\hbar}E_x t\right)\left(\left(q_x - \frac{2e}{\hbar}E_x t\right)^2 - 3\left(q_y - \frac{2e}{\hbar}E_y t\right)^2\right).\end{aligned} \qquad (13)$$

Here, $H$ is the generalized magnetic field (i.e., $H = B + \kappa M$) and $\alpha_3 = A_3\frac{\lambda_I \gamma}{T_c^2}$ originated from Ising SOC. The expectation value of the excess current density can be calculated by the time-dependent GL

equation with a stochastic force, namely

$$\hbar D\frac{\partial \Delta(\boldsymbol{r}, t)}{\partial t} = -\alpha(\boldsymbol{r}, t)\Delta(\boldsymbol{r}, t) + f(\boldsymbol{r}, t), \qquad (14)$$

where $D$ is the damping term for $\Delta(\boldsymbol{r}, t)$ resulting from the superconducting fluctuation, and $f(\boldsymbol{r}, t)$ is a stochastic force which generates $\langle|\Delta(\boldsymbol{q})|^2\rangle = \frac{k_B T}{\alpha(\boldsymbol{q})}$. The solution of the time-dependent GL equation is given by

$$\Delta(\boldsymbol{q}, t) = \frac{1}{\hbar D}\int_{-\infty}^t dt' f(\boldsymbol{q}, t')\exp\left(-\frac{1}{\hbar D}\int_{t'}^t d\tau\,\alpha(\boldsymbol{q}, \tau)\right), \qquad (15)$$

which also satisfies the boundary condition $\alpha(\boldsymbol{q}, \infty) = 0$. Then, the expectation value of the order parameter is

$$\left\langle|\Delta(\boldsymbol{q}, t)|^2\right\rangle = \frac{2k_B T}{\hbar D}\int_{-\infty}^t dt' f(\boldsymbol{q}, t')\exp\left(-\frac{1}{\hbar D}\int_{t'}^t d\tau\,\alpha(\boldsymbol{q}, \tau)\right). \qquad (16)$$

In analytical mechanics $\boldsymbol{A}$ can enters through the free energy (More generally, the Lagrangian). Considering infinitesimal variations $\delta\boldsymbol{A}$, we get $\delta F = -\int d\boldsymbol{r}\boldsymbol{J} \cdot \delta\boldsymbol{A}$, an expression used to obtain the current density $\boldsymbol{J}$. Thus, the current density operator is expressed as

$$\boldsymbol{J}(t) = -\frac{\delta F}{\Omega\delta\boldsymbol{A}} = \frac{1}{\Omega}\sum_q\left(-\frac{\partial\alpha(\boldsymbol{q}, t)}{\partial\boldsymbol{A}}\right)|\Delta(\boldsymbol{q}, t)|^2 = \frac{1}{\Omega}\sum_q\boldsymbol{J}(\boldsymbol{q}, t)|\Delta(\boldsymbol{q}, t)|^2 \qquad (17)$$

with $\Omega$ is the volume of the system. The expectation value of the current density is

$$\boldsymbol{J}(t) = \frac{2k_B T_c}{\Omega\hbar D}\sum_q\int_{-\infty}^t dt'\boldsymbol{J}(\boldsymbol{q}, t')\exp\left(-\frac{1}{\hbar D}\int_{t'}^t d\tau\,\alpha(\boldsymbol{q}, \tau)\right). \qquad (18)$$

By applying Eqs. (13), (18), we can obtain the conductivity as

$$\boldsymbol{J} = \sigma_1\boldsymbol{E} + \sigma_2 F(\boldsymbol{E}), \qquad (19)$$

where $\sigma_1 = \frac{e^2}{16\hbar}\left(\frac{T_c}{T - T_c}\right)$ and $\sigma_2 = -\frac{\pi e^3 m\alpha_3(H + o(H^3))}{64\hbar^3 k_B T_c}\left(\frac{T_c}{T - T_c}\right)^2$. In this context, $F(\boldsymbol{E}) = (E_x^2 - E_y^2, -2E_x E_y)$ denotes the nonlinear term of the in-plane electric field. As expected, $\sigma_2$ encapsulates the second harmonic signal, which is prominent near the superconducting transition where the small superconducting gap dominates the electrical transport.

We then performed the second harmonic measurement[3,53] with sweeping the perpendicular magnetic field at 4.5 K, which corresponds to the superconducting transition regime, for both magnetization "UP" and "DOWN" states. To distinguish the linear and nonlinear component, both the first ($R^\omega$) and second harmonic signals ($R^{2\omega}$) of the longitudinal magnetoresistance (Supplementary Fig. 6) are measured using a lock-in amplifier and applying the a.c. excitation current with an amplitude of 8 μA. The superconducting state fades off by increasing the magnetic field, manifested as the magnetoresistance in Supplementary Fig. 6a. We note that the magnetoresistance dip can be shifted to the left or right when the magnetization is reversed from "UP" to "DOWN" state, again indicating the presence of the proximity-induced exchange field $\kappa M$ from the magnetization of FGT. We then extracted the value of $\kappa M = \pm 10$mT, which is consistent with the critical field $B_C$ at which the $\Delta I_c$ vanishes for magnetization "UP" and "DOWN" state (Fig. 1e and Fig. 1g). In addition, the result that the $R^{2\omega}$ is antisymmetric with respect to the perpendicular magnetic field and magnetization (Supplementary Fig. 6b), i.e., $R^{2\omega}(-B_z, M_{DOWN}) = -R^{2\omega}(B_z, M_{UP})$, which is consistent with the magneto-toroidal nonreciprocal effect. Notably, the second harmonic

signal decays fast as the external magnetic field further increases, and the fermi energy becomes the dominant energy scale. This phenomenon confirms the important role of trigonal warping effect on the nonreciprocal superconducting transport in our PAIS device.

## Temperature dependence of nonreciprocal superconducting transport

We swept the direct current and monitored the change in resistance under various temperatures ranging from 1.6 to 3.5 K when fixing the magnetization "DOWN" state in a PAIS device with 7-layer NbSe$_2$ (Supplementary Fig. 7a). For each temperature, we observed significant nonreciprocal supercurrent with polarity "+" state (represented as $\Delta I_c < 0$). To further clarify the temperature dependence of this nonreciprocal effect, the temperature dependence of nonreciprocal efficiency $|\eta| = 2\left|\frac{I_c^+ + I_c^-}{I_c^+ - I_c^-}\right|$ are clearly plotted in Supplementary Fig. 7b. With increasing the temperature, this nonreciprocal efficiency $|\eta|$ decreases and its temperature dependence is well fitted by a $\sqrt{1 - \frac{T}{T_c}}$ function (the critical temperature $T_c = 4.8$K see in Supplementary Fig. 7c), consistent with the theoretic analysis (see Methods: Nonreciprocal critical current originated from finite momentum).

## Current-induced spin polarization

The coexistence of Ising SOC and trigonal warping at the Fermi surface can also facilitate the generation of current-induced perpendicular spin polarization. Specifically speaking, the electric field generated from the charge current would change the trigonally warped Fermi surfaces of spin-up and spin-down branches in both K and K' valleys and thus lift the valley degeneracy. This valley population imbalance would generate a z-spin polarization (the inset of Fig. 3e), which could produce a spin torque at the vdW interface and thus switch the magnetization in the PAIS material (Supplementary Fig. 8). To further elucidate the current-induced z-spin polarization, we calculated the current contribution to spin polarization using the Boltzmann equation[34,54] as follows.

The effective Hamiltonian of a monolayer NbSe$_2$ can be written as

$$H_{eff}(\boldsymbol{k}) = \frac{\hbar^2 \boldsymbol{k}^2}{2m} + \gamma k_x \left(k_x^2 - 3k_y^2\right)\tau_3, \tag{20}$$

where $\tau_z = \pm 1$ represent the valley degrees of freedom. Straightforward diagonalization of this Hamiltonian gives the eigenvalues

$$\varepsilon_\pm(\boldsymbol{k}) = \frac{\hbar^2 \boldsymbol{k}^2}{2m} \pm \gamma k_x \left(k_x^2 - 3k_y^2\right), \tag{21}$$

and eigenvectors

$$\psi_{\boldsymbol{k},+} = \begin{pmatrix} 1 \\ 0 \end{pmatrix} e^{i\boldsymbol{k}\cdot\boldsymbol{r}},$$

$$\psi_{\boldsymbol{k},-} = \begin{pmatrix} 0 \\ 1 \end{pmatrix} e^{i\boldsymbol{k}\cdot\boldsymbol{r}}.$$

For simplicity, we consider a line in the Brillouin zone along the $k_x$ direction (i.e., $k_y$ is a good quantum number). Solving the equation $\varepsilon_\pm(k_{F,\pm}) = \varepsilon_F$ to first order in $\gamma$ gives

$$k_{F,\pm} \approx k_F \left(1 \mp \frac{m\gamma k_F}{\hbar^2}\right) \equiv k_F(1 \mp \xi), \tag{22}$$

with $\xi = \frac{m\gamma k_F}{\hbar^2}$. Introducing an external electric field $\mathcal{E}$ could displace the Fermi surfaces by an amount $\Delta \boldsymbol{k}_\pm = -\frac{e\mathcal{E}\tau_\pm}{\hbar}$, where $\tau_\pm$ are the different

energy-dependent scattering rates. Generally, one can assume that $\tau_\pm = \tau(1 \pm \xi)$ with $\tau$ the relaxation time of the free-electron gas[55].

Then we calculate analytically the current contribution from each subband using the Boltzmann equation[34,54], one can obtain

$$\boldsymbol{J}_\pm = -e \int \boldsymbol{v}_{k,\pm} \frac{\partial f_{\boldsymbol{k},\pm}}{\partial \varepsilon} e v_{k,\pm} \tau_\pm \cdot \mathcal{E} d\boldsymbol{k} = \frac{e^2 \tau_\pm}{4\pi^2 \hbar} \iint \frac{\boldsymbol{v}_{k,\pm}}{v_{k,\pm}} \boldsymbol{v}_{k,\pm} \cdot \mathcal{E} dS_{F_\pm}, \tag{23}$$

where $f_{\boldsymbol{k},\pm}$ and $S_{F_\pm}$ are the electron distribution function and Fermi surfaces for the $\pm$ bands, respectively. By choosing $\mathcal{E} = \mathcal{E}\hat{\boldsymbol{x}}$ and assuming $v_{F,\pm} = v_F$, one can further get

$$J_{x,\pm} = \frac{e^2 \tau_\pm \mathcal{E}}{4\pi^2 \hbar} \int_0^{2\pi} v_F \cos^2\phi \, k_{F,\pm} \, d\phi = \frac{e^2 \mathcal{E}}{4\pi\hbar} v_F k_{F,\pm} \tau_\pm, \tag{24}$$

where we apply the relation $\boldsymbol{k} = k(\cos\phi, \sin\phi, 0)$. Thus, the total current density is given by

$$J = J_{x,+} + J_{x,-} = \frac{e^2 \mathcal{E}}{2\pi\hbar} v_F k_F \tau \left(1 + \xi^2\right). \tag{25}$$

The spin expectation value reads

$$\langle \boldsymbol{S} \rangle_{k,\pm} = \langle \psi_{k,\pm} | \boldsymbol{S} | \psi_{k,\pm} \rangle = \begin{pmatrix} 0 \\ 0 \\ \pm 1 \end{pmatrix}. \tag{26}$$

Therefore, the spin density can be calculated in an analogous way as

$$\langle \delta \boldsymbol{S} \rangle_\pm = \int \langle \delta \boldsymbol{S} \rangle_\pm \frac{\partial f}{\partial \varepsilon} e v_{k,\pm} \tau_\pm \cdot \mathcal{E} d\boldsymbol{k} = \mp \frac{e\mathcal{E}}{2\pi\hbar} k_{F,\pm} \tau_\pm \hat{\boldsymbol{z}}, \tag{27}$$

which yields the total spin density

$$\langle \delta \boldsymbol{S} \rangle = \langle \delta \boldsymbol{S} \rangle_+ + \langle \delta \boldsymbol{S} \rangle_- = \frac{2e\mathcal{E}}{\pi\hbar} k_F \tau \xi \hat{\boldsymbol{z}}. \tag{28}$$

Consequently, from the two Eq. (25) (28), we can obtain

$$\langle \delta \boldsymbol{S} \rangle \approx \frac{4m^2 \gamma}{eh^3} J\hat{\boldsymbol{z}}. \tag{29}$$

Here, we set $\gamma = 6.09 \times 10^{-21}$ J m$^3$ based on the previous literature[28] and obtain the relation of current induced spin density shown in Fig. 3e.

## Data availability

The data that support the findings of this study have been presented in the paper and the Supplementary Information. All source data can be acquired from the corresponding authors upon request. Source data are provided with this paper.

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

## Acknowledgements

This work was supported in part by the National Key R&D Program of China under Grant 2023YFF1203600 (S.-J.L.), the National Natural Science Foundation of China (12322407 (B.C.), 62122036 (S.-J.L.), 62034004 (F.M.), 61921005 (F.M.), 12074176 (B.C.)), the National Key R&D Program of China under Grant 2023YFF0718400 (B.C.), the Leading-edge Technology Program of Jiangsu Natural Science Foundation (BK20232004 (F.M.)), the Strategic Priority Research Program of the Chinese Academy of Sciences (XDB44000000 (F.M.)). F.M. and S.-J.L. would like to acknowledge support from the AIQ Foundation. J.Z. would like to acknowledge support from the National Natural Science

Foundation of China (623B1025). The microfabrication center of the National Laboratory of Solid State Microstructures (NLSSM) is acknowledged for their technique support. Crystal growth at Rutgers was supported by the center for Quantum Materials Synthesis (cQMS), funded by the Gordon and Betty Moore Foundation's EPiQS initiative through grant GBMF6402, and by Rutgers University.

## Author contributions

F.M., B.C. and S.-J.L. conceived the idea and supervised the whole project. J.-L.X. fabricated few-layer devices and performed transport measurements. J.X. carried out theoretical analysis and calculations. Y.D., X.-Y.C., L.W., Z.L. and J.Z. assist the measurements. J.-L.X., J.X., B.C., and S.-J.L. analyzed the data. N.W. and X.-H.C. grew NbSe$_2$ bulk crystals. X.X. and S.-W.C. grew Fe$_3$GeTe$_2$ bulk crystals. J.-L.X., J.X., B.C., S.-J.L. and F.M. wrote the manuscript with input from all authors.

## Competing interests

The authors declare no competing interests.
