## [Peer Review File · Nature Communications]

Electrical switching of Ising-superconducting nonreciprocity for quantum neuronal transistorREVIEWER COMMENTS

Reviewer #1 (Remarks to the Author):

Xiong et al., reported field-free electrical switching of Ising-superconducting diode effect in an artificial material FGT/NbSe₂ heterostructure and then utilized this phenomenon to demonstrate a proof-of-concept quantum neuronal transistor able to implement XOR gate function. Nonreciprocal quantum transport, e.g., superconducting diode effect, has been widely studied in condensed matter physics in recent years. Although this physics concept shows promising application, there has no work to report any possible device technology based on this effect. In this work, Xiong et al., stacked perpendicular-anisotropy ferromagnetic material Fe₃GeTe₂ flake and Ising spin-orbit coupling material NbSe₂ flake to fabricate high-quality heterostructure. Under different configurations of magnetization and current-flowing, they observed nonreciprocal superconductivity (or superconducting diode effect) in the fabricated device. Moreover, they demonstrated that the polarity of the nonreciprocal superconductivity (or superconducting diode effect) can be electrically switched by flowing current into the device to reverse the magnetization without the requirement of external magnetic field. This finding represents a major milestone towards future ultralow-power electronic device applications using superconducting diode effect. Finally, using the field-free electrical switching of superconducting diode effect, they propose a novel device concept, i.e., quantum neural transistor. Such transistor not only exhibits an excellent on-off ratio of 10³, but also allows for implementation of the well-known XOR gate function within the single device. This device demonstration in this work is strikingly interesting and important, because it not only provides the first example of exploiting superconducting diode effect for technological application but also breaks the conventional wisdom that the XOR gate function cannot be achieved in single device. In summary, this work presents interesting and significant experimental results, and also develops convincing theoretical model to well explain the experimental phenomena observed. I am happy to recommend this manuscript for publication in Nature Communications after the following questions are addressed.

1. To my best knowledge, it is the first time in this paper to report the implementation of XOR gate within one single device, which conventionally requires the use of multilayered networks. To highlight this point, I suggest the authors to add relevant discussion (where applicable) in the manuscript to point it clearly for the readership of interest.
2. The device shows an excellent on-off ratio, i.e., 10³. Such switching on and off is realized based on the variation of magnetoresistance. Compared to previously reported spintronic device, how large this magnetoresistance change is? Relevant discussion is suggested to add to emphasize the advantage of this device over previous technologies.
3. The superconductivity was demonstrated in NbSe₂ flake through the resistance-temperature characteristic. To further characterize the superconductive property, magnetic field dependence of superconductivity should be provided.
4. There is a trade-off between resistance area product and the magnetoresistance for the traditional magnetic tunnel junction-based spintronic devices. Different from the conventional spintronic devices, it seems that the proposed operating mechanism of the device can break such well-know trade-off. Authors are suggested to add relevant discussion to highlight the advantage over the previous device technology.

Reviewer #2 (Remarks to the Author):

Superconducting diode effect (SDE) is a promising phenomenon for understanding various aspects of superconductivity and for modelling superconducting technologies with both classical and quantum circuitry. While magnetic switching of nonreciprocal supercurrent is inherent to the unconventional superconductors with finite momentum Cooper pairing and/or finite magnetochiral anisotropy, electrical switching is desired for devising energy-efficient superconducting technologies. In the present manuscript (NCOMMS-24-07395-T), entitled "Electrical switching of Ising-superconducting nonreciprocity for quantum neuronal transistor", authors reported magnetization-determined intrinsic SDE, deterministic electrical switching of nonreciprocal

superconductivity, and a proof-of-concept nonreciprocal quantum neuronal transistor in a Fe₃GeTe₂/NbSe₂ van der Waals (vdW) heterostructure. The experimental results are explicitly supported by various theoretical calculations: the magnetization of sample is characterized by the Hall resistance as a function of the perpendicular magnetic field for various temperatures and the magnetization switching is demonstrated through sign reversal of Hall resistance by employing a current pulse. Association of magnetization-determined nonreciprocal superconductivity with finite-momentum Cooper pairing is reiterated by generalized Ginzburg-Landau theory and second harmonic calculations/measurements by adopting time-dependent Ginzburg-Landau equation. Current-induced spin polarization is also calculated by using the Boltzmann equation for monolayer NbSe₂ in which Ising spin-orbit interaction coexists with trigonal warping at the Fermi surface.

Though the magnetization-determined intrinsic SDE has already been demonstrated in several spin-orbit coupled superconductors, deterministic electrical switching of nonreciprocal superconductivity and a proof-of-concept nonreciprocal quantum neuronal transistor are exciting results for the emergent superconducting technologies. These novel aspects of SDE, induced by magnetoelectric effects, make the manuscript suitable for publication in "Nature Communications". The manuscript is well-written and the results are explicitly presented/supported, following minor clarifications would be of great help to grasp the underlying symmetry mechanisms quickly:

\textbf{1.} In a perpendicular-anisotropy Ising-superconducting (PAIS) quantum material, magnetization and magnetoelectric effects drive following two key results: (A) valley-contrasting Ising SOI and out-of-plane magnetization leads to SDE, characterized by finite-momentum Cooper pairing and/or finite magnetochiral anisotropy. (B) Current-controlled magnetization and thus SDE switching is associated with the breaking/lowering of C_3 symmetry, which could be accompanied by (i) in-plane electric polarization and/or (ii) trigonal warping at the Fermi surface. The association of valley-magnetization with electric polarization and warping effect allows magnetization switching via applied current. With this description at hand, is it feasible to further clarify following points?

\textbf{1.1} Is the crystallographic orientation of the PAIS material known? To be specific, in the given device setting with current along \hat{x} and magnetization along \hat{z} , does lowering of C_3 symmetry breaks mirror symmetry M_y or M_x ? Is current I_x flowing along armchair or zigzag direction?

\textbf{1.2} With broken M_y , both magnetochiral anisotropy and magneto-toroidal nonreciprocal directional dichroism (NDD) effect would be maximal, respectively, due to an optimized triple product $\hat{y} \cdot (M_z \times I_x)$ (i.e., maximum magnetochiral anisotropy) and a cross product $(P_y \times I_x)$ (i.e., maximum valley magnetization for in-plane electric polarization $\boldsymbol{P} = P_y$). On the other hand, with broken M_x , both magnetochiral anisotropy and the magneto-toroidal NDD effect would vanish. Does a similar symmetry argument also apply to optimizing the asymmetric trigonal warping effect? Answer to this question may help to understand which one of these two effects (magneto-toroidal NDD and warping) dominates or both can coexist and contribute simultaneously. Exact crystallographic orientation would also help to understand whether measured diode efficiency η (up to 13% at 1.6 K, as reported) is maximal or it can be further enhanced.

\textbf{2.} Are these critical temperature values $T_c \approx 2.95$ K (Fig. 1c) and $T_c \approx 4.8$ K (Extended Data Fig. 4c) measured under different bias current?

\textbf{3.} The proof-of-concept nonreciprocal quantum neuronal transistor is an exciting application of proposed electrical switching. The authors may find it interesting to mention a recently reported current-controlled switching of SDE in a Josephson junction made of chiral magnets: "Josephson transistor from the superconducting diode effect in domain wall and skyrmion magnetic racetracks [Richard Hess et al., Phys.Rev.B 108,174516,2023]".

Reviewer #3 (Remarks to the Author):

The authors report a magnetic field-free superconducting diode effect in Ising superconductor NbSe₂ by interfacing a ferromagnetic FGT having perpendicular anisotropy. The experimental results clearly showed electrical switching of the superconducting diode depending on remanent magnetization directions, and suitable device approach for XOR function. The authors further have explained possible mechanism and provided theoretical calculation supporting the experimental result. While the presented results are interesting, I hesitate to recommend to publish current manuscript since I feel further improvement is necessary to highlight this manuscript. In particular the field-free superconducting diode effect has been reported in various systems [Nat. Rev. Phys. 5, 558, 2023] and similar approach in Ising type superconductor/perpendicular anisotropy magnet of NbSe₂/CrPS₄ was reported but without field-free [Phys. Rev. Res. 5, L022064, 2023]. I have comments listed below.

(1) I don't agree the sentences in the abstract and introduction-"However, it remains yet to be achieved.", "but related progress is still absent."-which will give misleading information to readers. As the authors cited [Ref 15], many researchers have been reporting field-free superconducting diode in diverse systems.

(2) If there is magnetic proximity between NbSe₂ and FGT, the critical temperature change would be the fundamental feature to be shown, by comparing NbSe₂/FGT and FGT.

(3) In Fig.1e,g, the authors provided 4 data points each figure and it seems a linear relation between the current and field. However if it is associated with magnetic proximity from FGT, I would expect certain feature stemming from saturated magnetization of FGT (e.g. constant critical currents in a certain range of a magnetic field and its inversion depending on the magnetization UP and DOWN). Can the authors provide more data points?

(4) Could the authors explain in detail about the Hall signal (R_{xy}) in Fig.2a? What is the origin of this switching by current? Is this signal coming from the magnetized normal state NbSe₂ channel or spin-orbit torque in FGT channel? I am wondering this is relevant since the current range (mA) is far beyond the critical current shown in other figures.

(5) The authors provided the result from only one device. It is necessary to provide more device results to clarify the mechanism and to show functionality in the same way for device applications. For instance, as mentioned in the manuscript the mechanism seems intricate in this system. Another element needs to be considered is the number of NbSe₂ layer. The odd layer of NbSe₂ is known to have spatial symmetry breaking but not in the even layer [Ref 7], which is necessary to explore to insist the mechanism provided.

(6) The abbreviation "FGT" should be defined in the manuscript.

(7) In my opinion "on/off ratio" in the superconducting system is unsuitable. In principle it should be infinite based on zero resistance, but the ratio can be evaluated due to the lowest resolution of equipment which varies on each lab. Even though this ratio is important factor for reliability on device application but it is meaningless in superconducting devices.

Response to referees' comments

Reviewer #1 (Remarks to the Author):

Xiong et al., reported field-free electrical switching of Ising-superconducting diode effect in an artificial material FGT/NbSe₂ heterostructure and then utilized this phenomenon to demonstrate a proof-of-concept quantum neuronal transistor able to implement XOR gate function. Nonreciprocal quantum transport, e.g., superconducting diode effect, has been widely studied in condensed matter physics in recent years. Although this physics concept shows promising application, there has no work to report any possible device technology based on this effect. In this work, Xiong et al., stacked perpendicular-anisotropy ferromagnetic material Fe₃GeTe₂ flake and Ising spin-orbit coupling material NbSe₂ flake to fabricate high-quality heterostructure. Under different configurations of magnetization and current-flowing, they observed nonreciprocal superconductivity (or superconducting diode effect) in the fabricated device. Moreover, they demonstrated that the polarity of the nonreciprocal superconductivity (or superconducting diode effect) can be electrically switched by flowing current into the device to reverse the magnetization without the requirement of external magnetic field. This finding represents a major milestone towards future ultralow-power electronic device applications using superconducting diode effect. Finally, using the field-free electrical switching of superconducting diode effect, they propose a novel device concept, i.e., quantum neural transistor. Such transistor not only exhibits an excellent on-off ratio of 10^3 , but also allows for implementation of the well-known XOR gate function within the single device. This device demonstration in this work is strikingly interesting and important, because it not only provides the first example of exploiting superconducting diode effect for technological application but also breaks the conventional wisdom that the XOR gate function cannot be achieved in single device. In summary, this work presents interesting and significant experimental results, and also develops convincing theoretical model to well explain the experimental phenomena observed. I am happy to recommend this manuscript for publication in Nature Communications after the following questions are addressed.

Response: We thank the referee for highly appreciating the novelty and importance of our work, and recommending it for publication. By carrying out new experiments and adding new discussions in the revised manuscript, we have fully addressed all the comments raised by the referee. Below are detailed point-by-point replies to all comments.

1. To my best knowledge, it is the first time in this paper to report the implementation of XOR gate within one single device, which conventionally requires the use of multilayered networks. To highlight this point, I suggest the authors to add relevant discussion (where applicable) in the manuscript to point it clearly for the readership of interest.

Response: We thank the referee for the constructive suggestion. By following the reviewer's suggestion, we have added discussions in the revised manuscript to clarify the advantages of our device and highlight its significance. For ease of reviewing, the revision has been provided below, which is highlighted in yellow.

“As shown in Fig. 4g, when the input state and polarity state are set to (0,1) and (1,0), logic state “1” corresponding to a high resistance can be generated. By contrast, logic state “0” corresponding to a low resistance can be output when the input state and polarity state are set to (0,0) and (1,1). These results indicate that an XOR gate can be realized in a single quantum transistor. Note that this nonlinear Boole logic function cannot be implemented with a single traditional device and is conventionally thought to require multilayered networks⁴¹⁻⁴³.” [Lines 17-23, Page 8]

Reference:

“41. Xia, Q. & Yang, J. J. Memristive crossbar arrays for brain-inspired computing. Nat. Mater. 18, 309-323 (2019).

42. Kumar, S., Williams, R. S. & Wang, Z. Third-order nanocircuit elements for neuromorphic engineering. Nature 585, 518-523 (2020).

43. Sebastian, A., Le Gallo, M., Khaddam-Aljameh, R. & Eleftheriou, E. Memory devices and applications for in-memory computing. Nat. Nanotechnol. 15, 529-544 (2020).”

2. The device shows an excellent on-off ratio, i.e., 10^3 . Such switching on and off is realized based on the variation of magnetoresistance. Compared to previously reported spintronic device, how large this magnetoresistance change is? Relevant discussion is suggested to add to emphasize the advantage of this device over previous technologies.

Response: We thank the referee for the insightful suggestion. Basically, the magnetoresistance is defined as $MR = (R_H - R_L)/R_L$, where R_H and R_L represents the high resistance and low resistance, respectively. In our device, R_L is smaller than 0.004Ω , whereas R_H is about 9.8Ω , giving rise to a MR value of over 200,000%. This value is two orders of magnitude larger than that of conventional MgO-based MTJs [Nat. Mater. 3, 868 (2004); Nat. Mater. 3, 862 (2004); Appl. Phys. Lett. 93, 082508 (2008)] and one order of magnitude larger than the state-of-the-art value (19,000%) under similar experimental conditions [Science 360, 1214 (2018)]. To reflect the reviewer's suggestion, we have added relevant discussions in the revised manuscript to highlight the advantage of our device over previous technologies. For ease of reviewing, the revision has been provided below, which is highlighted in yellow.

“For the probe current I_{probe} of $+39 \mu A$, the ON-state resistance (R_{ON}) corresponding to the polarity “+” state is smaller than 0.004Ω , whereas the OFF-state resistance (R_{OFF}) corresponding to the polarity “-” state is about 9.8Ω , giving rise to an on/off ratio of 10^3 . This value is two orders of magnitude larger than that of MgO-based conventional MTJs³⁶⁻³⁸ and one order of magnitude larger than the state-of-the-art value under similar experimental conditions³⁹. Notably, this ratio depends on the lowest

resolution of the measurement system and can be further improved by improving the detection precision.” [Line 24, Page 7- Line 1, Page 8]

Reference:

“36. Parkin, S. S. P. et al. Giant tunnelling magnetoresistance at room temperature with MgO (100) tunnel barriers. *Nat. Mater.* 3, 862-867 (2004).

37. Yuasa, S. et al. Giant room-temperature magnetoresistance in single-crystal Fe/MgO/Fe magnetic tunnel junctions. *Nat. Mater.* 3, 868-871 (2004).

38. Ikeda, S. et al. Tunnel magnetoresistance of 604% at 300K by suppression of Ta diffusion in CoFeB/MgO/CoFeB pseudo-spin-valves annealed at high temperature. *Appl. Phys. Lett.* 93, 082508 (2008).

39. Song, T. et al. Giant tunneling magnetoresistance in spin-filter van der Waals heterostructures. *Science* 360, 1214-1218 (2018).”

3. The superconductivity was demonstrated in NbSe₂ flake through the resistance-temperature characteristic. To further characterize the superconductive property, magnetic field dependence of superconductivity should be provided.

Response: We thank the referee for this helpful suggestion. Following this suggestion, we have carried out additional experiments to characterize magnetic field dependence of superconductivity, with the corresponding results shown in Fig. R1. To reflect the referee’s suggestion, we have added the Fig. R1 into the supplementary materials [See Supplementary Fig. 3].

Fig. R1. Magnetic field and temperature dependence of the PAIS device resistance. **a**, Temperature dependence of the device resistance for magnetic fields ranging from 0 to 2.5 T. **b**, Magnetic field dependence of the device resistance for temperatures ranging from 2 to 5 K. This PAIS device consisting of seven-layer NbSe₂ flake was measured by applying an electrical current of 0.5 μ A.

4. There is a trade-off between resistance area product and the magnetoresistance for the traditional magnetic tunnel junction-based spintronic devices. Different from the conventional spintronic devices, it seems that the proposed operating mechanism of the

device can break such well-known trade-off. Authors are suggested to add relevant discussion to highlight the advantage over the previous device technology.

Response: We thank the referee for this constructive suggestion. As pointed out by the referee, the proposed operating mechanism of our PAIS device allows the simultaneous achievement of the giant magnetoresistance ($MR > 200,000\%$) and ultralow resistance area product ($RA \approx 0.1 \Omega \cdot \mu\text{m}^2$), which is inaccessible in conventional MTJs but urgently required for ultrahigh-density electronic applications [Nat. Electron. 3, 446 (2020); Nat. Electron. 6, 185 (2023)], as summarized in Fig. R2. To reflect the reviewer's suggestion, we have added Fig. R2 and related discussions to the supplementary materials [see Supplementary Fig. 12] to highlight the advantage of such operating mechanism in the revised manuscript.

Fig. R2. Comparison of magnetoresistance (MR) and resistance-area (RA) product between this work and previous literatures. Relationships between MR and RA of magnetic tunnel junctions (MTJs) with MgO-based (square)¹⁻³, AlO-based (circle)⁴, MgAlO-based (triangle)⁵ and van der Waals (vdW) material-based (hexagon)^{6,7} barrier are obtained from previous literatures. Different colors are used to distinguish different magnetic material components. The results from our work are denoted by the star symbol. The green solid line represents the tradeoff between MR and RA in the conventional MTJ. The RA of MTJ increases dramatically with the promotion of magnetoresistance.

For ease of reviewing, the revision has been provided below, which is highlighted in yellow.

“These results indicate that an XOR gate can be realized in a single quantum transistor. Note that this nonlinear Boole logic function cannot be implemented with a single traditional device and is conventionally thought to require multilayered networks. In addition, the PAIS device allows the simultaneous achievement of the giant on/off ratio ($>200,000\%$) and ultralow resistance area product ($\approx 0.1 \Omega \cdot \mu\text{m}^2$), which cannot be

realized in conventional MTJs (see detailed comparison in Supplementary Fig. 12) but is desirable for ultrahigh-density electronic applications^{44,45}.” [Lines 20-27, Page 8].

References:

“44. Dieny, B. et al. Opportunities and challenges for spintronics in the microelectronics industry. *Nat. Electron.* 3, 446-459 (2020).

45. Alam, S., Hossain, M. S., Srinivasa, S. R. & Aziz, A. Cryogenic memory technologies. *Nat. Electron.* 6, 185-198 (2023).”

Supplementary Reference:

“1. Parkin, S. S. P. et al. Giant tunnelling magnetoresistance at room temperature with MgO (100) tunnel barriers. *Nat. Mater.* 3, 862-867 (2004).

2. Yuasa, S. et al. Giant room-temperature magnetoresistance in single-crystal Fe/MgO/Fe magnetic tunnel junctions. *Nat. Mater.* 3, 868-871 (2004).

3. Ikeda, S. et al. Tunnel magnetoresistance of 604% at 300K by suppression of Ta diffusion in CoFeB/MgO/CoFeB pseudo-spin-valves annealed at high temperature. *Appl. Phys. Lett.* 93, 082508 (2008).

4. Wei, H. X. et al. 80% tunneling magnetoresistance at room temperature for thin Al-O barrier magnetic tunnel junction with CoFeB as free and reference layers. *J. Appl. Phys.* 101, 09B501 (2007).

5. Sukegawa, H. et al. Tunnel magnetoresistance with improved bias voltage dependence in lattice-matched Fe/spinel MgAl₂O₄/Fe(001) junctions. *Appl. Phys. Lett.* 96, 212505 (2010).

6. Song, T. et al. Giant tunneling magnetoresistance in spin-filter van der Waals heterostructures. *Science* 360, 1214-1218 (2018).

7. Wang, Z. et al. Tunneling Spin Valves Based on Fe₃GeTe₂/hBN/Fe₃GeTe₂ van der Waals Heterostructures. *Nano Lett.* 18, 4303-4308 (2018).”

Reviewer #2 (Remarks to the Author)

Superconducting diode effect (SDE) is a promising phenomenon for understanding various aspects of superconductivity and for modelling superconducting technologies with both classical and quantum circuitry. While magnetic switching of nonreciprocal supercurrent is inherent to the unconventional superconductors with finite momentum Cooper pairing and/or finite magnetochiral anisotropy, electrical switching is desired for devising energy-efficient superconducting technologies. In the present manuscript (NCOMMS-24-07395-T), entitled “Electrical switching of Ising-superconducting nonreciprocity for quantum neuronal transistor”, authors reported magnetization-determined intrinsic SDE, deterministic electrical switching of nonreciprocal superconductivity, and a proof-of-concept nonreciprocal quantum neuronal transistor in a $\text{Fe}_3\text{GeTe}_2/\text{NbSe}_2$ van der Waals (vdW) heterostructure. The experimental results are explicitly supported by various theoretical calculations: the magnetization of sample is characterized by the Hall resistance as a function of the perpendicular magnetic field for various temperatures and the magnetization switching is demonstrated through sign reversal of Hall resistance by employing a current pulse. Association of magnetization-determined nonreciprocal superconductivity with finite-momentum Cooper pairing is reiterated by generalized Ginzburg-Landau theory and second harmonic calculations/measurements by adopting time-dependent Ginzburg-Landau equation. Current-induced spin polarization is also calculated by using the Boltzmann equation for monolayer NbSe_2 in which Ising spin-orbit interaction coexists with trigonal warping at the Fermi surface.

Though the magnetization-determined intrinsic SDE has already been demonstrated in several spin-orbit coupled superconductors, deterministic electrical switching of nonreciprocal superconductivity and a proof-of-concept nonreciprocal quantum neuronal transistor are exciting results for the emergent superconducting technologies. These novel aspects of SDE, induced by magnetoelectric effects, make the manuscript suitable for publication in “Nature Communications”. The manuscript is well-written and the results are explicitly presented/supported, following minor clarifications would be of great help to grasp the underlying symmetry mechanisms quickly:

Response: We thank the referee for highly appreciating the novelty of our work, and recommending it for publication. By following the reviewer’s suggestions, we have added new discussions in the revised manuscript and fully addressed all the questions raised by the referee. The point-by-point responses are listed below.

1. In a perpendicular-anisotropy Ising-superconducting (PAIS) quantum material, magnetization and magnetoelectric effects drive following two key results: (A) valley-contrasting Ising SOI and out-of-plane magnetization leads to SDE, characterized by finite-momentum Cooper pairing and/or finite magnetochiral anisotropy. (B) Current-controlled magnetization and thus SDE switching is associated with the

breaking/lowering of C_3 symmetry, which could be accompanied by (i) in-plane electric polarization and/or (ii) trigonal warping at the Fermi surface. The association of valley-magnetization with electric polarization and warping effect allows magnetization switching via applied current. With this description at hand, is it feasible to further clarify following points?

Response: We thank the referee for the insightful and constructive comments. To reflect the referee’s suggestion, we have clarified all the questions raised by the referee in a point-by-point manner, as shown below.

1.1 Is the crystallographic orientation of the PAIS material known? To be specific, in the given device setting with current along \hat{x} and magnetization along \hat{z} , does lowering of C_3 symmetry breaks mirror symmetry M_y or M_x ? Is current I_x flowing along armchair or zigzag direction?

Response: We thank the referee for the comments. In our experiments, the crystallographic orientation of the NbSe₂ flake was confirmed by measuring the co-polarized SHG intensity as a function of relative angle between laser polarization and crystal orientation. As shown in Fig. R3, the maximum (minimum) intensity corresponds to the armchair (zigzag) direction of the NbSe₂ crystal. After confirming the crystallographic orientation, we designed the device geometry to set the direction of current along zigzag direction of the NbSe₂ sample.

Lowering of C_3 symmetry can indeed break mirror symmetry M_y in the given device setting with current along \hat{x} (zigzag direction) and magnetization along \hat{z} . Such symmetry breaking could result from the **ubiquitous strain and lattice mismatch at the vdW interface** and/or **trigonal warping effect**. First, the **ubiquitous strain and lattice mismatch** would break mirror symmetry M_y and generate the in-plane electric polarization P_y , as pointed out by the referee in the *Comment #1.2*. Second, with the current flowing along the zigzag direction, the Hamiltonian considering the **trigonal warping effect**, *i.e.*, $H = \left(\frac{k_x^2+k_y^2}{2m} - \mu\right)\sigma_0 + \lambda_1 k_x(k_x^2 - 3k_y^2)\sigma_z$, would break the mirror symmetry M_y since $\mathcal{M}_y H \mathcal{M}_y^{-1} \neq H$. In contrast, the mirror symmetry M_x is preserved since $\mathcal{M}_x H \mathcal{M}_x^{-1} = H$. Here, we adopted the equations $\mathcal{M}_x \sigma_z \mathcal{M}_x^{-1} = \mathcal{M}_y \sigma_z \mathcal{M}_y^{-1} = -\sigma_z$. **To reflect the reviewer’s suggestion, we have added the discussion on the characterization of crystallographic orientation in the revised main text and added the experimental results in Supplementary Materials [see Supplementary Fig. 14].** For ease of reviewing, the revision has been provided below, which is highlighted in yellow.

*“We mechanically exfoliated the NbSe₂ and FGT flakes onto a highly doped Si wafer covered by a 300-nm-thick SiO₂ layer. **Before the device fabrication, the crystallographic orientation of the NbSe₂ flake was characterized by measuring the co-polarized SHG intensity as a function of relative angle between laser polarization and crystal orientation. In the following fabrication process, we designed the device***

geometry to set the direction of current along zigzag direction of the NbSe₂ sample in the experiments. As shown in Supplementary Fig. 14, the maximum (minimum) intensity corresponds to the armchair (zigzag) direction of the crystal, confirming the current flowing along the zigzag direction. The thickness values of these flakes were identified with optical contrast and a Bruker MultiMode 8 atomic force microscope.” [Lines 7-15, Page 9]

“Here, κ is a constant to ensure dimensional consistency. The effective Hamiltonian $H(\mathbf{k})$ has the mirror symmetry \mathcal{M}_x since $\mathcal{M}_x H(\mathbf{k}) \mathcal{M}_x^{-1} = H(\mathbf{k})$, but breaks the mirror symmetry \mathcal{M}_y since $\mathcal{M}_y H(\mathbf{k}) \mathcal{M}_y^{-1} \neq H(\mathbf{k})$.” [Lines 6-9, Page 11]

Fig. R3. Second harmonic generation measurements to determine the relationship between the current direction and crystallographic orientation. **a**, Optical image of a PAIS device. The scale bar is 3 μm . **b**, Polar plot of the second harmonic generation (SHG) signal. The green arrow represents the current direction.

1.2 With broken M_y , both magnetochiral anisotropy and magneto-toroidal nonreciprocal directional dichroism (NDD) effect would be maximal, respectively, due to an optimized triple product $\hat{y} \cdot (\mathbf{M}_z \times \mathbf{I}_x)$ (i.e., maximum magnetochiral anisotropy) and a cross product $\mathbf{P}_y \times \mathbf{I}_x$ (i.e., maximum valley magnetization for in-plane electric polarization $\mathbf{P} = \mathbf{P}_y$). On the other hand, with broken M_x , both magnetochiral anisotropy and the magneto-toroidal NDD effect would vanish. Does a similar symmetry argument also apply to optimizing the asymmetric trigonal warping effect? Answer to this question may help to understand which one of these two effects (magneto-toroidal NDD and warping) dominates or both can coexist and contribute simultaneously. Exact crystallographic orientation would also help to understand whether measured diode efficiency η (up to 13% at 1.6 K, as reported) is maximal or it can be further enhanced.

Response: We thank the referee for the insightful comment and constructive suggestion. When considering the asymmetric trigonal warping effect, we can also confirm the condition of optimizing the superconducting diode and current-controlled magnetization effects based on the similar symmetry argument mentioned by the referee. Specifically, as the current direction is applied along the zigzag direction (i.e.,

x axis), the asymmetric trigonal warping effect can break the mirror symmetry M_y (see *Response #1.1*), which would give rise to the appearance of SDE and its electrical switching, as presented below in details.

First, when considering trigonal warping effect, we can obtain that the SDE could arise and has the similar symmetry dependence as that derived from the magnetochiral anisotropy effect. Specifically, with the assist of the magnetization proximity, the trigonal warping effect could break the M_y symmetry. Such M_y symmetry breaking facilitates an optimized finite momentum of Cooper pairs under zero external magnetic field by following the relation $q_x \propto \hat{x} \cdot (\mathbf{M}_z \times \hat{y})$ (see detailed derivations in the Appendix I: Finite momentum of Cooper pairs at the end of this response letter). The optimized finite momentum of Cooper pairs would lead to a magnetization (\mathbf{M}_z) determined superconducting diode effect (SDE) with maximum diode efficiency. This diode efficiency η follows the formula $\eta \propto \hat{x} \cdot (\mathbf{M}_z \times \hat{y}) \propto \hat{y} \cdot (\mathbf{M}_z \times \hat{x})$. Such formula ($\eta \propto \hat{y} \cdot (\mathbf{M}_z \times \hat{x})$) from trigonal warping effect and the formula $\eta \propto \hat{y} \cdot (\mathbf{M}_z \times \mathbf{I}_x)$ from magnetochiral anisotropy effect have the same dependence of M_y symmetry.

Second, when considering trigonal warping effect, we can infer a similar symmetry dependence of the electrical switching behaviors as that derived from the NDD effect. Basically, an electric field produced by the charge current would change the trigonally warped Fermi surfaces of spin-up and spin down branches for K and K' valleys and thus lifts the valley degeneracy. This valley population imbalance then generates a z-spin accumulation, which could produce a spin torque to switch the magnetization and the corresponding SDE. Such z-spin accumulation follows the formula $\langle \delta \mathbf{S} \rangle \propto \hat{y} \times \mathbf{J}_x$ (see detailed derivations in the Appendix II: Current-induced spin polarization at the end of response letter). Notably, this formula ($\langle \delta \mathbf{S} \rangle \propto \hat{y} \times \mathbf{I}_x$) based on trigonal warping effect and the formula ($\mathbf{P}_y \times \mathbf{I}_x$) based on magneto-toroidal nonreciprocal directional dichroism (NDD) effect have the same dependence of M_y symmetry.

Therefore, according to the symmetry argument, the magneto-toroidal NDD and trigonal warping effect are allowed to coexist and thus can simultaneously contribute to the emergence of SDE as well as the current-driven switching of SDE. Moreover, in our experiments, the current is set to flow along the zigzag direction, thus leading to a maximum efficiency of SDE and its electrical switching based on the above discussions. **To reflect the reviewer's comments, we have added new discussions on the symmetry mechanisms in the revised manuscript and Supplementary Materials.** For ease of reviewing, the revision has been provided below, which is highlighted in yellow.

"On the one hand, the ubiquitous strain and lattice mismatch at the vdW interface can break the M_y mirror symmetry of the NbSe₂, and lead to in-plane electrical polarization (\mathbf{P})." [Lines 26-28, Page 5]

"On the other hand, with the current along the zigzag direction, the M_y mirror symmetry of this system can also be broken due to the intricate interplay between valley-contrasting trigonal warping and the magnetic proximity effect (see more detailed discussion in Supplementary Materials)." [Lines 3-6, Page 6]

2. Are these critical temperature values $T_c \approx 2.95$ K (Fig. 1c) and $T_c \approx 4.8$ K (Extended Data Fig. 4c) measured under different bias current?

Response: We thank the referee for the helpful comment. The critical temperature values in Fig. 1c and Extended Data Fig. 4c are measured in two different devices under the same bias current (0.5 μ A). Despite the different critical temperature, these two devices exhibit similar behaviors of electrically switchable superconducting nonreciprocity (see Supplementary Fig. 9-11 in the revised Supplementary Materials), indicating the robustness of our device. Such difference of critical temperature in these two devices is attributed to the different sample thickness of the two devices (one with five-layer NbSe₂ ($T_c \approx 2.95$ K) and another with seven-layer NbSe₂ ($T_c \approx 4.8$ K)). To avoid any possible confusion, we have added statements in the revised manuscript to clarify that the experimental data are from different devices with distinct sample thickness. For ease of reviewing, the revision has been provided below, which is highlighted in yellow.

“We swept the direct current and monitored the change in resistance under various temperatures ranging from 1.6 to 3.5 K when fixing the magnetization “DOWN” state in a PAIS device with 7-layer NbSe₂ (Supplementary Fig. 7a).” [Lines 22-24, Page 15]
“It is noted that this electrically switchable nonreciprocal superconductivity is reproducible and observed in the both PAIS devices with odd-layer and even-layer NbSe₂ (see Section IX of the Supplementary Materials). Fully understanding of such electrical switching of superconducting nonreciprocity requires microscopic models considering the details of the magnetic proximity effect and the symmetry breaking at the vdW interface.” [Line 33, Page 6-Line 6, Page 7]

3. The proof-of-concept nonreciprocal quantum neuronal transistor is an exciting application of proposed electrical switching. The authors may find it interesting to mention a recently reported current-controlled switching of SDE in a Josephson junction made of chiral magnets: "Josephson transistor from the superconducting diode effect in domain wall and skyrmion magnetic racetracks [Richard Hess et al., Phys.Rev.B 108,174516,2023]".

Response: We thank the referee for bringing this important literature to our attention. To reflect the reviewer’s suggestion, we have added this reference into our revised manuscript (as new Ref. 32). For ease of reviewing, the revision has been provided below, which is highlighted in yellow.

“This value is two order of magnitude larger than that of MgO-based conventional MTJs and one order of magnitude larger than the state-of-the-art value (1,9000%) under similar experimental conditions. Notably, this ratio depends on the lowest resolution of the measurement system and can be further improved by improving the detection precision. We also note that a similar superconducting transistor has been theoretically proposed by the current-controlled switching of superconducting

nonreciprocity in a Josephson junction with chiral magnet⁴⁰.” [Line 27, Page 7-Line 4, Page 8]

Reference:

“40. Hess, R., Legg, H. F., Loss, D. & Klinovaja, J. Josephson transistor from the superconducting diode effect in domain wall and skyrmion magnetic racetracks. *Phys. Rev. B* 108, 174516 (2023).”

Reviewer #3 (Remarks to the Author)

The authors report a magnetic field-free superconducting diode effect in Ising superconductor NbSe₂ by interfacing a ferromagnetic FGT having perpendicular anisotropy. The experimental results clearly showed electrical switching of the superconducting diode depending on remanent magnetization directions, and suitable device approach for XOR function. The authors further have explained possible mechanism and provided theoretical calculation supporting the experimental result. While the presented results are interesting, I hesitate to recommend to publish current manuscript since I feel further improvement is necessary to highlight this manuscript. In particular the field-free superconducting diode effect has been reported in various systems [Nat. Rev. Phys. 5, 558, 2023] and similar approach in Ising type superconductor/perpendicular anisotropy magnet of NbSe₂/CrPS₄ was reported but without field-free [Phys. Rev. Res. 5, L022064, 2023]. I have comments listed below.

Response: We thank the reviewer for the positive evaluation that our result is interesting. The studies of the superconducting diode effect (SDE) have attracted extensive attention due to its unprecedented potential for developing next-generation quantum device [Nat. Rev. Phys. 5, 558 (2023)]. Although the SDE has been reported in several systems, including NbSe₂/CrPS₄ as mentioned by the referee, the switching of SDE in these systems can be only realized with the assistant of the external magnetic field. The requirement of external magnetic field hinders the exploitation of SDE for designing low-power consumption and quantum electronic devices suitable for ultrahigh-density integration. In our work, we for the first time realize the field-free electrical switching of nonreciprocal Ising superconductivity, and give the first example of neuromorphic quantum transistor utilizing such capability of electrical switching. These main advances in our work are summarized below:

- (1) In this work, we for the first time experimentally achieved field-free electrical switching of SDE by constructing the van der Waals heterostructure with the intricate interplay between Ising-type superconductivity and magnetic proximity. Our proposed electrical switching technology breaks through the limitation of external magnetic field for ultrahigh density integration and makes a key step towards the development of nonreciprocal quantum electronic devices.
- (2) We proposed a proof-of-concept neuromorphic transistor based on the on/off switching of SDE. This neuromorphic quantum transistor can implement the XOR logic gate and faithfully emulate biological functionality of a cortical neuron. Notably, such device functionality is inaccessible with the previous SDE systems mentioned by the referee and conventional semiconductor devices. Our work thus opens up a promising avenue for developing emergent quantum electronic devices based on nonreciprocal quantum transport.

Thereby, our work has both conceptual and technique advances over previous works which only focus on physical mechanisms of SDE.

(1) I don't agree the sentences in the abstract and introduction-"However, it remains yet to be achieved.", "but related progress is still absent."-which will give misleading information to readers. As the authors cited [Ref 15], many researchers have been reporting field-free superconducting diode in diverse systems.

Response: We thank the referee for this comment. We meant to use this sentence in our original manuscript to refer to the fact that the **field-free electrical switching** of superconducting diode effect remains yet to be achieved. **To avoid misleading the readers, we have modified this statement in the revised manuscript.** For ease of reviewing, the revision has been provided below, which is highlighted in yellow.

*“Realizing electrical switching of the polarity of the nonreciprocal transport without the assistant of external magnetic field is essential to the development of nonreciprocal quantum devices. **However, electrical switching of superconducting nonreciprocity remains yet to be achieved.**” [Lines 24-27, Page 1]*

*“A key step for that purpose is achieving electrical switching of the nonreciprocal superconductivity without the assistant of external magnetic field, **but the electrically switchable nonreciprocal superconductivity is still absent.**” [Lines 22-24, Page 2]*

(2) If there is magnetic proximity between NbSe₂ and FGT, the critical temperature change would be the fundamental feature to be shown, by comparing NbSe₂/FGT and FGT.

Response: We thank the referee for the helpful suggestion. Following this suggestion, we have carried out additional experiments to characterize temperature dependence of resistances in the NbSe₂, FGT and NbSe₂/FGT samples, with the corresponding results shown in Fig. R4. The superconducting transition temperature T_c of the NbSe₂/FGT heterostructure with 7-layer (7L) NbSe₂ is 4.8 K at zero field. This value (4.8 K) is lower than that of the 7L-NbSe₂ sample (6.1 K), which can be attributed to the pair-breaking effect induced by the magnetization of FGT. **To reflect the referee's suggestion, we have added the Fig. R4 into the supplementary materials [See Supplementary Fig. 4].**

Fig. R4. Temperature dependent normalized longitudinal resistances (defined as $R_{xx}(T)/R_{xx}(8K)$) of the NbSe₂ (green line), FGT (black line) and NbSe₂/FGT (orange line) samples.

(3) In Fig. 1e, g, the authors provided 4 data points each figure and it seems a linear relation between the current and field. However if it is associated with magnetic proximity from FGT, I would expect certain feature stemming from saturated magnetization of FGT (e.g. constant critical currents in a certain range of a magnetic field and its inversion depending on the magnetization UP and DOWN). Can the authors provide more data points?

Response: We thank the referee for the insightful comment. To follow the referee's suggestion, we have characterized the superconducting diode effect (SDE) under larger external magnetic fields (i.e., -100 mT, -50 mT, 50 mT and 100 mT) for magnetization UP (Fig. R5) and DOWN (Fig. R6) states. We extracted the nonreciprocal supercurrents ΔI_c at large external magnetic fields and presented these new results in the revised Fig. 1e, g (also see Fig. R7). We observe that the nonreciprocal supercurrent ΔI_c increases linearly as we gradually increase the magnetic field, while ΔI_c is decreased as the external magnetic field is beyond a certain breakdown field. Such observation can be attributed to the fact that the external magnetic field would enhance the superconducting nonreciprocity and this field would also suppress the superconductivity itself. As we include the magnetic proximity effect in this intuitive picture, the total effective external magnetic field should be the sum of the applied external magnetic field and the effective field induced by proximity magnetization. In this sense, a bias would be introduced to the field dependence of the SDE, thus resulting in a zero-field superconducting nonreciprocity. To reflect the referee's suggestion, we have added more discussions on magnetic field dependence of superconducting nonreciprocity and added new results at large external magnetic fields in the revised Fig. 1e, g. For ease of reviewing, the revision has been provided below, which is highlighted in yellow.

"This nonreciprocity with "+" polarity is retained until $B_z = -10$ mT and eventually reversed to that with "-" polarity (i.e., $\Delta I_c < 0$). When the magnetic field further increases, the nonreciprocal supercurrent ΔI_c is then suppressed beyond certain magnetic field due to field-induced breakdown of superconductivity (Supplementary Fig. 5)." [Lines 9-13, Page 4]

Fig. R5. Current dependences of the resistance under different perpendicular magnetic fields $B=-100$ mT, -50 mT, 50 mT, and 100 mT for both positive and negative currents at 1.6 K when the magnetization is set as “UP” state.

Fig. R6. Current dependences of the resistance under different perpendicular magnetic fields $B=-100$ mT, -50 mT, 50 mT, and 100 mT for both positive and negative currents at 1.6 K when the magnetization is set as “DOWN” state.

Fig. R7. Magnetization-determined nonreciprocal Ising superconductivity under different perpendicular magnetic fields. **a**, The nonreciprocal component of the critical current ΔI_c as a function of the magnetic field for the magnetization “UP” state. **b**, The nonreciprocal component of the critical current ΔI_c as a function of the magnetic field for the magnetization “DOWN” state.

(4) Could the authors explain in detail about the Hall signal (R_{xy}) in Fig.2a? What is the origin of this switching by current? Is this signal coming from the magnetized normal state NbSe₂ channel or spin-orbit torque in FGT channel? I am wondering this is relevant since the current range (mA) is far beyond the critical current shown in other figures.

Response: We thank the referee for the insightful comments. As pointed out by the referee, the probe current ($500 \mu\text{A}$) is indeed beyond the critical current I_c . This would

break the superconductivity and thus enable the normal charge current to flow into the FGT channel. Thereby, we can conclude that the Hall signal (R_{xy}) in Fig.2a (also see Fig. R8) should mainly result from the anomalous hall effect induced by the perpendicular magnetization in the FGT layer. In addition, the magnetized NbSe₂ channel induced by magnetic proximity of FGT can also contribute to a small fraction of the measured hall resistance. In this sense, the sign change of the Hall resistance induced by current pulses indicates the current-driven switching of perpendicular magnetization.

We attribute this electrical switching of perpendicular magnetization to the current-induced z-spin polarization in the PAIS device. This z-spin polarization will generate a spin torque via spin-exchange coupling and thus reverse the perpendicular magnetization. Such magnetization reversal leads to the switching of the sign of the Hall resistance (which is manifested at a probe current beyond superconducting critical current I_c) and the switching of superconducting nonreciprocity when the probe current is smaller than I_c . To reflect the reviewer's comments, we have added the related theoretical analysis in *Section: Current-induced spin polarization* in the main text of revised manuscript.

Fig. R8. Current-induced magnetization switching at zero field at 1.6 K. The sign of Hall resistance is reversed by sweeping the pulsed current.

(5) The authors provided the result from only one device. It is necessary to provide more device results to clarify the mechanism and to show functionality in the same way for device applications. For instance, as mentioned in the manuscript the mechanism seems intricate in this system. Another element needs to be considered is the number of NbSe₂ layer. The odd layer of NbSe₂ is known to have spatial symmetry breaking but not in the even layer [Ref 7], which is necessary to explore to insist the mechanism provided.

Response: We thank the referee for the constructive suggestion. Following this suggestion, we have carried out new experiments and presented the corresponding electrical transport data on total 3 devices in the revised manuscript. These 3 devices with different thickness of NbSe₂ are labeled as device #1 (with 5-layer NbSe₂), device #2 (with 7-layer NbSe₂) and device #3 (with 6-layer NbSe₂), respectively. The corresponding results measured on these three different devices are shown in Fig. R9, Fig. R10 and Fig. R11, respectively. We observed that these devices with both even-layer NbSe₂ and odd-layer NbSe₂ exhibit similar behaviors of electrically switchable

superconducting diode effect and functionality of nonreciprocal neural transistor. These observations indicate excellent reproducibility of our experimental results. Note that the supercurrent diode effect is not theoretically expected to occur in even-layer NbSe₂, owing to the restored inversion symmetry pointed out by the referee. The reasons for generating the supercurrent diode effect in even-layer device can be attributed to the following two aspects: 1) The crystal symmetry could be broken by the ubiquitous strain and lattice mismatch at the vdW interface. 2) The relatively weak electronic coupling between the layers makes the physical properties of few-layer NbSe₂ similar as the monolayers, which is consistent with the mechanism proposed for the emergence of external field induced SDE in both odd-layer and even-layer NbSe₂ samples in ref. 7.

To reflect the reviewer’s suggestions, we have added the new experimental results and relevant discussion in the revised manuscript and Supplementary Materials [see Section IX of the Supplementary Materials]. For ease of reviewing, we have provided the revision below, which is highlighted in yellow.

“This current-induced z-spin could also produce a spin torque at the vdW interface and thus contribute to the switching of the magnetization in the PAIS material (Extended Data Fig. 5). **It is noted that this electrically switchable nonreciprocal superconductivity is reproducible and observed in the both PAIS devices with odd-layer and even-layer NbSe₂ (see Section IX of the Supplementary Materials).** Fully understanding of such electrical switching of superconducting nonreciprocity requires microscopic models considering the details of the magnetic proximity effect and the symmetry breaking at the vdW interface.” [Line 31, Page 6-Line 6, Page 7]

Device #1 with five-layer NbSe₂

Fig. R9. Electrically switchable superconducting nonreciprocity and functionality of nonreciprocal neural transistor in a five-layer device. **a**, Current-induced magnetization switching at zero field at 1.6 K. **b**, electrically switchable nonreciprocal superconducting transport. **c**, Deterministic switching by a series of current pulses applied in the device. The width and magnitude of the current pulses are 200 μs and 8 mA, respectively. The resistance is measured by using a small d.c. excitation current of +33 μA . **d**, The responses of spike to the input current pulses for the polarity “+” and “-” states. **e**, The XOR function in the nonreciprocal neural transistor. The dashed boxes represent the logic states for input and polarity combinations (0,1), (1,1), (0,0) and (1,0), respectively.

Fig. R10. Electrically switchable superconducting nonreciprocity and functionality of nonreciprocal neural transistor in a seven-layer device. **a**, Current-induced magnetization switching at zero field at 1.6 K. **b**, electrically switchable nonreciprocal superconducting transport. **c**, Deterministic switching by a series of current pulses applied in the device. The width and magnitude of the current pulses are 200 μs and 10 mA, respectively. The resistance is measured by using a small d.c. excitation current of +39 μA . **d**, The responses of spike to the input current pulses for the polarity “+” and “-” states. **e**, The XOR function in the nonreciprocal neural transistor. The dashed boxes represent the logic states for input and polarity combinations (0,1), (1,1), (0,0) and (1,0), respectively.

Fig. R11. Electrically switchable superconducting nonreciprocity and functionality of nonreciprocal neural transistor in a six-layer device. **a**, Current-induced magnetization switching at zero field at 1.6 K. **b**, electrically switchable nonreciprocal superconducting transport. **c**, Deterministic switching by a series of current pulses applied in the device. The width and magnitude of the current pulses are 200 μs and 12 mA, respectively. The resistance is measured by using a small d.c. excitation current of +36 μA . **d**, The responses of spike to the input current pulses for the polarity “+” and “-” states. **e**, The XOR function in the nonreciprocal neural transistor. The dashed boxes represent the logic states for input and polarity combinations (0,1), (1,1), (0,0) and (1,0), respectively.

(6) The abbreviation "FGT" should be defined in the manuscript.

Response: We thank the referee for the helpful suggestion. Following the reviewer’s suggestion, we have defined the abbreviation “FGT” in the revised manuscript. For ease of reviewing, we have provided the revision below, which is highlighted in yellow.

“The PAIS device was fabricated by stacking vdW magnet Fe_3GeTe_2 (FGT) and Ising superconductor 2H-NbSe2 (Fig. 1a and 1b).” [Lines 17-18, Page 3]

(7) In my opinion "on/off ratio" in the superconducting system is unsuitable. In principle it should be infinite based on zero resistance, but the ratio can be evaluated due to the lowest resolution of equipment which varies on each lab. Even though this

ratio is important factor for reliability on device application but it is meaningless in superconducting devices.

Response: We thank the referee for the comment. As a transistor, an on/off ratio is required. Analog to the conventional semiconductor transistors, we defined the similar parameter “on/off ratio” for our proposed quantum transistor in our work. Although this ratio depends on the detection precision of the measurement system, which might vary on each lab, the defined ratio could help evaluate the potential of our proposed transistor in the field of neuromorphic computing. To reflect the reviewer’s concern, we have removed the on/off ratio appeared in the abstract of our original manuscript and modified the relevant description about the on/off ratio, to avoid misleading the readership of interest. For ease of reviewing, we have provided the revision below, which is highlighted in yellow.

“By taking advantage of this field-free electric switching of superconducting nonreciprocity, we demonstrate a proof-of-concept nonreciprocal quantum neuronal transistor.” [Lines 2-4, Page 2]

“Notably, this ratio depends on the lowest resolution of the measurement system and can be further improved by improving the detection precision.” [Line 30, Page 7- Line 1, Page 8]

Appendix I: Finite momentum of Cooper pairs

To derive the finite momentum of Cooper pairs induced by the trigonal warping effect, we employ the generalized Ginzburg-Landau (GL) theory, where the free energy density as a functional of the superconducting order Δ parameter reads

$$f_s(\Delta, \mathbf{q}) = [\alpha_0 + \alpha_2 \mathbf{q}^2 + \alpha_3 q_x (q_x^2 - 3q_y^2) B_z^{eff}] |\Delta|^2 + \frac{\beta}{2} |\Delta|^4,$$

where $\alpha_0 = A_0(T - T_c)$, $\alpha_2 = \frac{\hbar^2}{4m}$, $\alpha_3 = A_3 \frac{\lambda_I \gamma}{T_c^2}$ and $\beta > 0$ with $A_0, A_3 > 0$ are numerical constants. Here, λ_I and γ are model parameters which depend on the effective Hamiltonian with trigonal warping effect.

For simplicity, we express the above free energy density in a more compact form, $f_s(\Delta, \mathbf{q}) = \alpha(\mathbf{q}) |\Delta|^2 + \frac{\beta}{2} |\Delta|^4$, with $\alpha(\mathbf{q}) = \alpha_0 + \alpha_2 \mathbf{q}^2 + \alpha_3 \mathbf{q}^3 \cos 3\theta B_z^{eff}$, where θ represents the angle between \mathbf{q} and q_x axis. By minimizing $\alpha(\mathbf{q})$ over \mathbf{q} , one can find the Cooper pair momentum \mathbf{q}_0 in the equilibrium state. With current flowing along the zigzag direction, this current $I = I \hat{\mathbf{x}}$ would change this Cooper pair momentum to $q_x = (\mathbf{q} - \mathbf{q}_0) \cdot \hat{\mathbf{x}}$, then $\alpha(q_x) = \alpha_0 + \alpha_2 q_x^2 + \alpha_3 q_x^3 \hat{\mathbf{x}} \cdot (\mathbf{B}_z^{eff} \times \hat{\mathbf{y}})$. Then, we find the Cooper pair momentum in the equilibrium state by minimizing $\alpha(q_x)$ over q_x , that is

$$q_x^0 = \frac{2\alpha_0}{3\alpha_3} \hat{\mathbf{x}} \cdot (\mathbf{B}_z^{eff} \times \hat{\mathbf{y}}),$$

which is a direct result of the mirror symmetry \mathcal{M}_y breaking.

Appendix II: Current-induced spin polarization

To elucidate the coexistence of Ising SOC and trigonal warping at the Fermi surface can facilitate the generation of current-induced perpendicular spin polarization, we start with the effective Hamiltonian of a monolayer NbSe₂, which can be written as

$$H_{eff}(\mathbf{k}) = \frac{\hbar^2 \mathbf{k}^2}{2m} + \gamma k_x (k_x^2 - 3k_y^2) \tau_3,$$

where $\tau_z = \pm 1$ represent the valley degrees of freedom. Straightforward diagonalization of this Hamiltonian gives the eigenvalues

$$\varepsilon_{\pm}(k) = \frac{\hbar^2 k^2}{2m} \pm \gamma k_x (k_x^2 - 3k_y^2),$$

and eigenvectors

$$\begin{aligned}\psi_{k,+} &= \begin{pmatrix} 1 \\ 0 \end{pmatrix} e^{ikr}, \\ \psi_{k,-} &= \begin{pmatrix} 0 \\ 1 \end{pmatrix} e^{ikr}.\end{aligned}$$

Introducing an external electric field \mathcal{E} could displace the Fermi surfaces (*i.e.*, $\varepsilon_{\pm}(k_{F,\pm}) = \varepsilon_F$) by an amount $\Delta \mathbf{k}_{\pm} = -\frac{e\mathcal{E}\tau_{\pm}}{\hbar}$, where the Fermi momentum $k_{F,\pm}$ can be given as $k_{F,\pm} \approx k_F \left(1 \mp \frac{m\gamma k_F}{\hbar^2}\right) \equiv k_F(1 \mp \xi)$, and τ_{\pm} are the different energy-dependent scattering rates. Generally, one can assume that $\tau_{\pm} = \tau(1 \pm \xi)$ with τ the relaxation time of the free-electron gas.

Then we calculate analytically the current contribution from each subband using the Boltzmann equation, one can get

$$J_{\pm} = -e \int \mathbf{v}_{k,\pm} \frac{\partial f_{k,\pm}}{\partial \varepsilon} e v_{k,\pm} \tau_{\pm} \cdot \mathcal{E} d\mathbf{k} = \frac{e^2 \tau_{\pm}}{4\pi^2 \hbar} \iint \frac{\mathbf{v}_{k,\pm}}{v_{k,\pm}} \mathbf{v}_{k,\pm} \cdot \mathcal{E} dS_{F_{\pm}},$$

where $f_{k,\pm}$ and $S_{F_{\pm}}$ are the electron distribution function and Fermi surfaces for the \pm bands, respectively. By choosing $\mathcal{E} = \mathcal{E}\hat{\mathbf{x}}$ and assuming $v_{F,\pm} = v_F$, one can further get

$$J_{x,\pm} = \frac{e^2 \tau_{\pm} \mathcal{E}}{4\pi^2 \hbar} \int_0^{2\pi} v_F \cos^2 \phi k_{F,\pm} d\phi = \frac{e^2 \mathcal{E}}{4\pi \hbar} v_F k_{F,\pm} \tau_{\pm},$$

where we apply the relation $\mathbf{k} = k(\cos \phi, \sin \phi, 0)$. Thus, the total current density is given by

$$J_x = J_{x,+} + J_{x,-} = \frac{e^2 \mathcal{E}}{2\pi \hbar} v_F k_F \tau (1 + \xi^2) \hat{\mathbf{x}}.$$

The spin expectation value reads

$$\langle \mathbf{S} \rangle_{k,\pm} = \langle \psi_{k,\pm} | \mathbf{S} | \psi_{k,\pm} \rangle = \begin{pmatrix} 0 \\ 0 \\ \pm 1 \end{pmatrix}.$$

A similar calculation can be performed to calculate the total spin density,

$$\langle \delta \mathbf{S} \rangle = \sum_{+,-} \int \langle \delta \mathbf{S} \rangle_{\pm} \frac{\partial f}{\partial \varepsilon} v_{k,\pm} \tau_{\pm} \cdot \boldsymbol{\varepsilon} d\mathbf{k} = \frac{2e\mathcal{E}}{\pi\hbar} k_F \tau \hat{\mathbf{z}}.$$

Consequently, the current density and the optimized spin density can establish such a relationship,

$$\langle \delta \mathbf{S} \rangle \approx -\frac{4m^2\gamma}{e\hbar^3} \hat{\mathbf{y}} \times \mathbf{J}_x.$$

REVIEWERS' COMMENTS

Reviewer #1 (Remarks to the Author):

The authors have addressed the questions from the reviewers. I would recommend the acceptance.

Reviewer #2 (Remarks to the Author):

I appreciate the changes and clarifications in the revised manuscript, including many guided by the comments of the Reviewer #1 and the Reviewer #3. The Authors addressed the remarks with satisfactory arguments. The manuscript, which was already well written, now has a better connection to symmetry constraints, physical phenomena, and experimental relevance of neuronal transistor. With these updates, the Authors have made the manuscript suitable for the broad readership. As this original work provides important advances towards electrical switching of nonreciprocal superconductivity and demonstrates proof-of-concept neuronal transistor able to implement XOR gate function, I recommend its publication in Nature communications.

Optional: The Authors may rewrite and strengthen the statement ``We also note that a similar superconducting transistor has been theoretically proposed by the current-controlled switching of superconducting nonreciprocity in a Josephson junction with chiral magnet⁴⁰." [Line 27, Page 7-Line 4, Page 8]". The phrase "similar superconducting transistor" could undermine the novelty of the proposed superconducting neuronal transistor. The mechanism of electrical switching in the present manuscript (spin-orbit torque intertwined with trigonal warping effect and/or magneto-toroidal nonreciprocal directional dichroism) is completely distinct from the electrical switching in the Ref. 40 (current-driven nontrivial magnetic texture such as domain walls and/or skyrmions in magnetic racetrack materials used as weak link). Spin-orbit torque could also be linked with nontrivial magnetic textures, but that is not the case in the Ref.40. Second, Authors may also highlight that, unlike electrical switching in semiconducting/superconducting hybrid Josephson junction transistor [Ref.40], electrical switching and the superconducting neuronal transistor concept is realizable in all-metallic junction-free bulk superconductors.

Reviewer #3 (Remarks to the Author):

The authors have addressed my concerns and comments adequately in the revised manuscript. I recommend its publication in Nature Communications.

Response to referees' comments

Reviewer #1 (Remarks to the Author):

The authors have addressed the questions from the reviewers. I would recommend the acceptance.

Response: We thank the referee for recommending the publication of our work in Nature Communications.

Reviewer #2 (Remarks to the Author):

I appreciate the changes and clarifications in the revised manuscript, including many guided by the comments of the Reviewer #1 and the Reviewer #3. The Authors addressed the remarks with satisfactory arguments. The manuscript, which was already well written, now has a better connection to symmetry constraints, physical phenomena, and experimental relevance of neuronal transistor. With these updates, the Authors have made the manuscript suitable for the broad readership. As this original work provides important advances towards electrical switching of nonreciprocal superconductivity and demonstrates proof-of-concept neuronal transistor able to implement XOR gate function, I recommend its publication in Nature communications.

Optional: The Authors may rewrite and strengthen the statement ``We also note that a similar superconducting transistor has been theoretically proposed by the current-controlled switching of superconducting nonreciprocity in a Josephson junction with chiral magnet⁴⁰." [Line 27, Page 7-Line 4, Page 8]". The phrase "similar superconducting transistor" could undermine the novelty of the proposed superconducting neuronal transistor. The mechanism of electrical switching in the present manuscript (spin-orbit torque intertwined with trigonal warping effect and/or magneto-toroidal nonreciprocal directional dichroism) is completely distinct from the electrical switching in the Ref. 40 (current-driven nontrivial magnetic texture such as domain walls and/or skyrmions in magnetic racetrack materials used as weak link).

Spin-orbit torque could also be linked with nontrivial magnetic textures, but that is not the case in the Ref.40. Second, Authors may also highlight that, unlike electrical switching in semiconducting/superconducting hybrid Josephson junction transistor [Ref.40], electrical switching and the superconducting neuronal transistor concept is realizable in all-metallic junction-free bulk superconductors.

Response: We thank the referee for highly appreciating the novelty of our work, and recommending the publication of our work in Nature Communications. By following the reviewer's suggestions, we have revised the statement and added new discussions in the revised manuscript. For ease of reviewing, we have provided the revisions in the following, which have been highlighted in yellow.

“We also note that a superconducting transistor based on the distinct physical mechanism has been theoretically proposed in a Josephson junction with chiral magnet. Unlike the Josephson transistor based on the junction structure in that work, our superconducting neuronal transistor is realized in all-metallic junction-free superconductors.” *[Line 24-29, Page 7]*

Reviewer #3 (Remarks to the Author):

The authors have addressed my concerns and comments adequately in the revised manuscript. I recommend its publication in Nature Communications.

Response: We thank the referee for recommending the publication of our work in Nature Communications.